# Sliced Denoising: A Physics-Informed Molecular Pre-Training Method

**Yuyan Ni[1,3]**[*][†]**, Shikun Feng [2]**[*]**, Weiying Ma[2], Zhiming Ma[1], Yanyan Lan[2,4]** [‡]

[1]Academy of Mathematics and Systems Science, Chinese Academy of Sciences
[2]Institute for AI Industry Research (AIR), Tsinghua University
[3]University of Chinese Academy of Sciences
[4]Beijing Academy of Artificial Intelligence

## Abstract

While molecular pre-training has shown great potential in enhancing drug discovery, the lack of a solid physical interpretation in current methods raises concerns about whether the learned representation truly captures the underlying explanatory factors in observed data, ultimately resulting in limited generalization and robustness. Although denoising methods offer a physical interpretation, their accuracy is often compromised by ad-hoc noise design, leading to inaccurate learned force fields. To address this limitation, this paper proposes a new method for molecular pre-training, called sliced denoising (SliDe), which is based on the classical mechanical intramolecular potential theory. SliDe utilizes a novel noise strategy that perturbs bond lengths, angles, and torsion angles to achieve better sampling over conformations. Additionally, it introduces a random slicing approach that circumvents the computationally expensive calculation of the Jacobian matrix, which is otherwise essential for estimating the force field. By aligning with physical principles, SliDe shows a 42% improvement in the accuracy of estimated force fields compared to current state-of-the-art denoising methods, and thus outperforms traditional baselines on various molecular property prediction tasks.[1]

## 1 Introduction

Molecular representation learning plays a crucial role in a variety of drug discovery tasks, including molecular property prediction (Schütt et al., 2018; 2021; Thölke & Fabritiis, 2022), molecular generation (Bilodeau et al., 2022; Jing et al., 2022), and protein-ligand binding (Gao et al., 2023; Zheng et al., 2019). To overcome the challenge of insufficient labeled data, various molecular pre-training methods have been proposed to obtain a universal molecular representation, including the contrastive approaches (Fang et al., 2022; Wang et al., 2022; Stärk et al., 2022; Li et al., 2022) and the predictive approaches (Rong et al., 2020; Fang et al., 2021; Zhu et al., 2022; Liu et al., 2023a).

According to Bengio et al. (2013), a good representation is often one that captures the posterior distribution of the underlying explanatory factors for the observed input data. Regarding molecular representation, we posit that an ideal representation must adhere to the underlying physical principles that can accurately and universally illustrate molecular patterns. However, the majority of existing pre-training methods draw inspiration from pre-training tasks in computer vision and natural language processing and thus overlook the underlying physical principles.

Nevertheless, designing self-supervised tasks that align with physical principles remains challenging. To the best of our knowledge, only one kind of the unsupervised molecular pre-training method has an explicit physical interpretation, i.e. the 3D denoising approach (Zaidi et al., 2023; Feng et al., 2023), which aims to learn an approximate force field for molecules. However, we have found that this approximate force field largely deviates from the true force field, due to inappropriate assumptions such as assuming a molecular force field is isotropic in coordinate denoising (Zaidi et al.,

---

[*]Equal contribution. [†] Work was done while Yuyan Ni was a research intern at AIR. [‡] Correspondence to `lanyanyan@air.tsinghua.edu.cn`.

[1]The code is released publicly at `https://github.com/fengshikun/SliDe`.

2023) or treating certain parts being isotropically in fractional denoising (Feng et al., 2023). Consequently, existing denoising methods still harbor a significant bias from physical laws, which can hinder downstream results, as depicted by the experiments conducted in Feng et al. (2023) and our own experiments in Appendix B.5. Therefore, it remains an essential issue to design a denoising method that better aligns with physical principles.

It should be noted that energy function is a pivotal factor in determining the quality of representation learning in denoising methods. Firstly, the Boltzmann distribution used for noise sampling, which determines the conformations on which the network learns its force field, is derived from the energy function. Secondly, the learned force field, which aims to align regression targets with the true molecular force field, is designated by the gradient of the energy function. As a result, a precise energy function facilitates the network to acquire accurate force fields for typical molecules, consequently enhancing the physical consistency of the representation.

Following the aforementioned analysis, we suggest utilizing the classical mechanical intramolecular potential energy function and approximating it in the quadratic form using relative coordinates, i.e. bond lengths, angles, and torsion angles, with certain parameters. Inspired by the previous theoretical findings that associate the quadratic energy function with a Gaussian distribution through the Boltzmann distribution, we then propose a novel noise strategy, called BAT noise. Specifically, BAT noise introduces Gaussian noise to bond lengths, angles, and torsion angles, and their respective variances are predetermined by parameters within the energy function. This approach allows BAT noise to better approximate the true molecular distribution when compared to other existing methods. The resulting conformations from our strategy are closer to common low-energy structures than previous approaches, providing an advantage for effective representation learning.

The objective of the denoising target is to regress the molecular force field, i.e. the gradient of the energy function w.r.t. Cartesian coordinates. However, the energy function is defined in relative coordinates, thus requiring a change of variables in the differential. Specifically, the gradient of the energy function in relation to relative coordinates is readily acquirable in the form of the by-term product of the BAT noise and the parameter. Applying a variable change requires estimation of the Jacobian matrix of the coordinate transformation function, which is nevertheless computationally expensive. To address this issue, we introduce a random slicing technique that converts the Jacobian estimation into simple operations of coordinate noise additions and BAT noise acquisitions.

Thus we have developed a novel and efficient method, known as sliced denoising (SliDe), which is equivalent to learning the force field of the utilized energy function. Consequently, SliDe possesses the ability to align better with physical principles by estimating a more precise force field. To facilitate the learning process, we introduce a Transformer-based network architecture that explicitly encodes relative coordinate information and generates equivariant atom-wise features tailored for the sliced denoising task. Our contributions are summarized as follows:

1) Methodologically, we suggest the use of physical consistency as a guiding principle for molecular representation learning, and under this principle, we develop a novel sliced denoising method and corresponding network architecture.

2) Theoretically, we derive BAT noise from the classical mechanical energy function and establish the equivalence between learning the force field and our sliced denoising method.

3) Experimentally, we demonstrate that SliDe outperforms existing pre-training methods in terms of physical consistency and downstream performance on QM9, MD17 and ANI-1x datasets.

## 2 BACKGROUND

Denoising is a kind of self-supervised learning task in molecular representation learning and has achieved outstanding results in many downstream tasks (Zhou et al., 2023; Feng et al., 2023; Zaidi et al., 2023; Luo et al., 2023; Liu et al., 2023b; Jiao et al., 2023). It refers to corrupting original molecules by specific noise and training the neural networks to predict the noise, thus reconstructing the molecules. Significant benefit of denoising over other pre-training methods is that it has been proven to be equivalent to learning a molecular force field, which is physically interpretable.

Coordinate denoising (Coord) (Zaidi et al., 2023) involves the addition of Gaussian noise to atomic coordinates of equilibrium structures, with subsequent training of the model to predict the noise

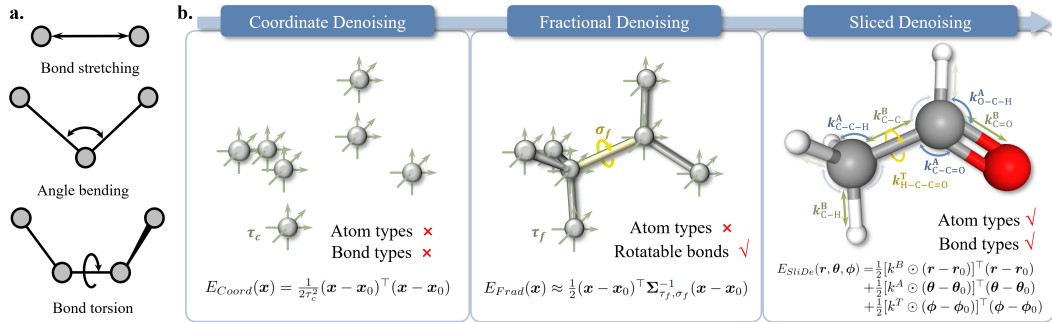

Figure 1: **a**. Illustrations of bond stretching, bond angle bending, and bond torsion interactions. **b**. Comparison of the three denoising methods in terms of energy functions. Coordinate denoising learns an isotropic energy in Cartesian coordinates that does not discriminate different atom types and bond types. Based on coordinate denoising, fractional denoising treats the rotatable bonds in special. In contrast, sliced denoising performs fine-grained treatment for different atom types and bond types, enabling the most physically consistent description of the molecule.

from the noisy input. They establish the equivalence between coordinate denoising and force field learning, under the assumption of isotropic Gaussian noise. For a given sampled molecule $\mathcal{M}$, perturb the equilibrium structure $\boldsymbol{x}_0$ by $p(\boldsymbol{x}|\boldsymbol{x}_0) \sim \mathcal{N}(\boldsymbol{x}_0, \tau_c^2 I_{3N})$, where $\boldsymbol{x}$ denotes the noisy conformation, $N$ denotes the number of atoms in the molecule, and $I_{3N}$ is the identity matrix of size $3N$, the subscript $c$ stands for the coordinate denoising approach. Assume the molecular distribution satisfies the energy-based Boltzmann distribution w.r.t the energy function $E_{Coord}$, then

$$\mathcal{L}_{Coord}(\mathcal{M}) = E_{p(\boldsymbol{x}|\boldsymbol{x}_0)p(\boldsymbol{x}_0)}||GNN_\theta(\boldsymbol{x}) - (\boldsymbol{x} - \boldsymbol{x}_0)||^2 \tag{1}$$

$$\simeq E_{p(\boldsymbol{x})}||GNN_\theta(\boldsymbol{x}) - (-\nabla_{\boldsymbol{x}} E_{Coord}(\boldsymbol{x}))||^2, \tag{2}$$

where $GNN_\theta(\boldsymbol{x})$ refers to a graph neural network with parameters $\theta$ that takes the conformation $\boldsymbol{x}$ as input and returns node-level predictions. The notation $\simeq$ indicates the equivalence between different optimization objectives for the GNN. The proof is supplemented in the appendix A.

To account for the anisotropic molecular distribution, fractional denoising (Frad) (Feng et al., 2023) proposes introducing a hybrid noise on the dihedral angles of rotatable bonds and atomic coordinates and fractionally denoising the coordinate noise. This specially designed denoising task allows for a physical interpretation of learning force field. For a given sampled molecule $\mathcal{M}$, the equilibrium structure $\boldsymbol{x}_0$ is perturbed by $p(\boldsymbol{\psi}_a|\boldsymbol{\psi}_0) \sim \mathcal{N}(\boldsymbol{\psi}_0, \sigma_f^2 I_m)$ and $p(\boldsymbol{x}|\boldsymbol{x}_a) \sim \mathcal{N}(\boldsymbol{x}_a, \tau_f^2 I_{3N})$, where $\boldsymbol{\psi}_a$ and $\boldsymbol{\psi}_0$ correpond to the dihedral angles of rotatable bonds in structures $\boldsymbol{x}_a$ and $\boldsymbol{x}_0$ respectively, with $m$ representing the number of rotatable bonds, and the subscript $f$ standing for Frad. Assume the molecular distribution satisfies the energy-based Boltzmann distribution w.r.t the energy function $E_{Frad}$, we have

$$\mathcal{L}_{Frad}(\mathcal{M}) = E_{p(\boldsymbol{x}|\boldsymbol{x}_a)p(\boldsymbol{x}_a|\boldsymbol{x}_0)p(\boldsymbol{x}_0)}||GNN_\theta(\boldsymbol{x}) - (\boldsymbol{x} - \boldsymbol{x}_a)||^2 \tag{3}$$

$$\simeq E_{p(\boldsymbol{x})}||GNN_\theta(\boldsymbol{x}) - (-\nabla_{\boldsymbol{x}} E_{Frad}(\boldsymbol{x}))||^2. \tag{4}$$

The proof is also supplemented in the appendix A. A summary of denoising pre-training methods is provided in Appendix D.1.

The aforementioned work has made efforts to learn physically interpretable molecular representations by designing noise distributions and their corresponding energy functions based on certain chemical priors. However, their energy function is coarse-grained as shown in Figure 1b., lacking the capability to capture highly complex interaction information, such as bond stretching, angle bending, and bond torsion composed of different bond types and atom types. In contrast, our noise distributions and force fields are derived from a classical mechanical energy function, which is more consistent with the characteristics of true molecules.

## 3 OUR APPROACH

Inspired by the aforementioned deduction, we can conclude that in pursuit of designing an effective and interpretable denoising pre-training task, physical consistency can be achieved by developing an energy function that accurately approximates the true molecular energy. This, in turn, leads to a

better noise distribution capable of sampling low-energy molecules, and a correspondingly improved force field learned through denoising. Following this guiding principle, we first establish a physical informed energy function in section 3.1, followed by the design of a noise distribution in section 3.2. In section 3.3, we present a denoising task aimed at learning the force field of the aforementioned energy function. Finally, in section 3.4, we introduce the network architecture developed for our denoising method.

## 3.1 Energy Function

According to classical molecular potential energy theory (Mol, 2020; Zhou & Liu, 2022), the total intramolecular potential energy can be attributed to five types of interactions: bond stretching, bond angle bending, bond torsion, electrostatic, and van der Waals interactions. Figure 1a. depicts the first three interactions. The energy function, in its general form, can be expressed as follows:

$$E(\boldsymbol{r}, \boldsymbol{\theta}, \boldsymbol{\phi}) = \frac{1}{2} \sum_{i \in \mathbb{B}} k_i^B (r_i - r_{i,0})^2 + \frac{1}{2} \sum_{i \in \mathbb{A}} k_i^A (\theta_i - \theta_{i,0})^2$$
$$+ \sum_{i \in \mathbb{T}} k_i^T (1 - cos(\omega_i(\phi_i - \phi_{i,0}))) + E_{elec} + E_{vdW}, \tag{5}$$

where $\boldsymbol{r}$, $\boldsymbol{\theta}$, and $\boldsymbol{\phi}$ represent vectors of the bond lengths, bond angles, and bond torsion angles of the molecule, respectively. The index $i$ corresponds to the element in the vector. $\boldsymbol{r}_0$, $\boldsymbol{\theta}_0$, and $\boldsymbol{\phi}_0$ correspond to the respective equilibrium values. The parameter vectors $\boldsymbol{k}^B$, $\boldsymbol{k}^A$, and $\boldsymbol{k}^T$ determine the interaction strength, while the parameter vectors $\boldsymbol{\omega}$ determine the torsion periodicity. The index set $\mathbb{B}$, $\mathbb{A}$, $\mathbb{T}$ correspond to the bonds, angles, and torsion angles in the molecule, respectively.

In order to approximate it as a quadratic form, which is often required to enable the equivalence based on previous proof, we put forward two approximation operations. Firstly, when $\boldsymbol{\phi} \to \boldsymbol{\phi}_0$, a Taylor expansion is utilized to express the bond torsion interaction in the quadratic form:

$$1 - cos(\omega_i(\phi_i - \phi_{i,0})) = 1 - [1 - \frac{1}{2}(\omega_i(\phi_i - \phi_{i,0}))^2 + o((\phi_i - \phi_{i,0})^2)] \approx \frac{1}{2}\omega_i^2(\phi_i - \phi_{i,0})^2.$$

The approximation is reasonable since the noise scale in denoising methods is usually small. Secondly, we drop the last two terms in order to get a quadratic form of energy function in equation 6. Despite these long-range electrostatic and van der Waals interactions are important in classical simulations, we find the accuracy of the approximated energy function is much higher than existing self-supervised pre-training methods in section 4.1.

$$E_{BAT}(\boldsymbol{r}, \boldsymbol{\theta}, \boldsymbol{\phi}) = \frac{1}{2} \sum_{i \in \mathbb{B}} k_i^B (r_i - r_{i,0})^2 + \frac{1}{2} \sum_{i \in \mathbb{A}} k_i^A (\theta_i - \theta_{i,0})^2 + \frac{1}{2} \sum_{i \in \mathbb{T}} k_i^T \omega_i^2 (\phi_i - \phi_{i,0})^2. \tag{6}$$

In Figure 1b., we compare our energy function with that of Coord and Frad. Their formulations provide a general outline of the energy function in an averaged manner using only one or two parameters. Unfortunately, they fail to capture the nuanced energetic characteristics of molecules. In contrast, our energy function carefully describes the impact of different atomic types and bond types on energy using specific parameters. Therefore, our approach is more closely aligned with the true physical properties of molecules.

## 3.2 Noise Design

With the common assumption that the conformation distribution of a molecule follows the Boltzmann distribution (Boltzmann, 1868), i.e. $p \propto exp(-E)$, we can derive the conformation distribution corresponding to our quadratic energy function.

$$p(\boldsymbol{r}, \boldsymbol{\theta}, \boldsymbol{\phi}) = \frac{1}{Z} exp(-E_{BAT}(\boldsymbol{r}, \boldsymbol{\theta}, \boldsymbol{\phi})) \tag{7}$$

$$= \prod_{i \in \mathbb{B}} \frac{1}{Z_i^B} exp(-k_i^B \frac{(r_i - r_{i,0})^2}{2}) \prod_{i \in \mathbb{A}} \frac{1}{Z_i^A} exp(-k_i^A \frac{(\theta_i - \theta_{i,0})^2}{2}) \prod_{i \in \mathbb{T}} \frac{1}{Z_i^T} exp(-k_i^T \omega_i^2 \frac{(\phi_i - \phi_{i,0})^2}{2}), \tag{8}$$

where $Z$, $Z_i^B$, $Z_i^A$, $Z_i^T$ are normalization factors. According to equation 8, the conformation distribution can be expressed as a joint distribution of independent Gaussian on bond lengths, bond angles, and torsion angles. Therefore we can outline the following noise strategy.

**Definition 3.1** (BAT noise). The BAT noise strategy refers to perturbing the equilibrium structure by adding independent Gaussian noise on every **b**ond length, **a**ngle and **t**orsion angle:

$$\boldsymbol{r} \sim \mathcal{N}(\boldsymbol{r}_0, diag(\frac{1}{\boldsymbol{k}^B})), \boldsymbol{\theta} \sim \mathcal{N}(\boldsymbol{\theta}_0, diag(\frac{1}{\boldsymbol{k}^A})), \boldsymbol{\phi} \sim \mathcal{N}(\boldsymbol{\phi}_0, diag(\frac{1}{\boldsymbol{k}^T \odot \boldsymbol{\omega}^2})), \tag{9}$$

where $\odot$ means multiply item by item, $diag(\cdot)$ represents a diagonal matrix whose diagonal elements are the elements of the vector. The variances are determined by the parameters that can be obtained in prior, such as from the parameter files of molecular simulation tools.

Detail implementations of the BAT noise can be found in Appendix C.1. Since $E_{BAT}$ approximates the true molecular energy function, the sampling distribution of BAT noise resembles the true molecular distribution. This guarantees realistic sampled conformations that are beneficial for learning effective representations.

## 3.3 SLICED DENOISING

Since the energy function is based on bond lengths, angles, and torsion angles, the gradient of the energy function can be represented as a simple form with respect to the relative coordinate:

$$\nabla_{\boldsymbol{d}} E_{BAT}(\boldsymbol{d}) = [\boldsymbol{k}^B \odot (\boldsymbol{r} - \boldsymbol{r}_0), \boldsymbol{k}^A \odot (\boldsymbol{\theta} - \boldsymbol{\theta}_0), \boldsymbol{k}^T \odot \boldsymbol{\omega}^2 \odot (\boldsymbol{\phi} - \boldsymbol{\phi}_0)]^\top, \tag{10}$$

where $\boldsymbol{d} = (\boldsymbol{r}, \boldsymbol{\theta}, \boldsymbol{\phi})$. However, we need to derive a simple expression for the gradient of our energy function with respect to Cartesian coordinates and ensure the learning of the force field of $E_{BAT}$ by minimizing

$$E_{p(\boldsymbol{x}|\boldsymbol{x}_0)}||GNN_\theta(\boldsymbol{x}) - \nabla_{\boldsymbol{x}} E_{BAT}(\boldsymbol{d}(\boldsymbol{x}))||^2. \tag{11}$$

For this purpose, we propose expanding the gradient using the chain rule and expressing the force field target as the gradient of the energy function with respect to the relative coordinates and a Jacobian matrix of the coordinate transformation. A rigorous formulation is presented as follows.

Firstly, we define a coordinate transformation function for a molecule $\mathcal{M}$ that maps from Cartesian coordinates to relative coordinates:

$$f^{\mathcal{M}} : \mathbb{R}^{3N} \longrightarrow (\mathbb{R}_{\geq 0})^{m_1} \times ([0, 2\pi))^{m_2} \times ([0, 2\pi))^{m_3} \tag{12}$$
$$\boldsymbol{x} \longmapsto \boldsymbol{d} = (\boldsymbol{r}, \boldsymbol{\theta}, \boldsymbol{\phi}),$$

where $m_1$, $m_2$, and $m_3$ are numbers of bonds, angles, and torsion angles respectively. The mapping is well-defined, as these values can be uniquely determined by the Cartesian coordinates. Although $\boldsymbol{\theta}$, $\boldsymbol{\phi}$ are defined on a torus, we can establish a homeomorphism between the Euclidean space $\mathbb{R}^{m_2+m_3}$ and $([0, 2\pi)\backslash\{p_i\})^{m_2} \times ([0, 2\pi)\backslash\{p_j\})^{m_3}$, where $p_i, p_j$ are any points in $[0, 2\pi), i = 1 \cdots m_2, j = 1 \cdots m_3$ (Zorich, 2016). As the denoising method only involves conformations in a small neighborhood $V$ around the equilibrium conformation $\boldsymbol{d}_0$, we can select $p_i, p_j$ such that $V \in ([0, 2\pi)\backslash\{p_i\})^{m_2} \times ([0, 2\pi)\backslash\{p_j\})^{m_3}$. Consequently, the coordinate transformation function defined on a neighborhood can be regarded as a mapping between the Euclidean spaces $\mathbb{R}^{3N} \longrightarrow \mathbb{R}^M$, where $M \triangleq m_1 + m_2 + m_3$.

Assume that the coordinate transformation function $f^{\mathcal{M}}$ is continuously differentiable with continuous partial derivatives. Then the force field can be expressed by

$$\nabla_{\boldsymbol{x}} E_{BAT}(f(\boldsymbol{x}))^\top = \nabla_{\boldsymbol{d}} E_{BAT}(\boldsymbol{d})^\top \cdot J(\boldsymbol{x}), \tag{13}$$

where $J(\boldsymbol{x}) = \begin{pmatrix} \frac{\partial f_1^{\mathcal{M}}(\boldsymbol{x})}{\partial x_1} & \cdots & \frac{\partial f_1^{\mathcal{M}}(\boldsymbol{x})}{\partial x_{3N}} \\ \vdots & & \vdots \\ \frac{\partial f_M^{\mathcal{M}}(\boldsymbol{x})}{\partial x_1} & \cdots & \frac{\partial f_M^{\mathcal{M}}(\boldsymbol{x})}{\partial x_{3N}} \end{pmatrix} \in \mathbb{R}^{M \times 3N}$ is the Jacobian matrix. Then the target in equation 11 can be written as

$$E_{p(\boldsymbol{x}|\boldsymbol{x}_0)}||GNN_\theta(\boldsymbol{x}) - \nabla_{\boldsymbol{d}} E_{BAT}(\boldsymbol{d})^\top \cdot J(\boldsymbol{x})||^2. \tag{14}$$

For each noisy conformation, the Jacobian matrix can be estimated via numerical differentiation, which is time-consuming. To efficiently learn the force field, we devise a cleverly designed asymptotically unbiased estimator that does not require the computation of the Jacobian matrix, by utilizing two computational techniques.

Firstly, a random slicing technique is introduced to estimate the target regression loss through the projection of the GNN and force field onto random vectors, as illustrated by lemma 3.2.

**Lemma 3.2** (Random Slicing). $\forall \boldsymbol{a}, \boldsymbol{b}, \boldsymbol{v} \in \mathbb{R}^{3N}$, $\sigma > 0$, $\boldsymbol{v} \sim \mathcal{N}(\boldsymbol{0}, \sigma^2 I_{3N})$, then

$$||\boldsymbol{a} - \boldsymbol{b}||^2 = \frac{1}{\sigma^2} E_{\boldsymbol{v}}[(\boldsymbol{a} - \boldsymbol{b})^\top \cdot \boldsymbol{v}]^2. \tag{15}$$

An intuitive illustration of lemma 3.2 is that the L2 norm of the vector $\boldsymbol{a} - \boldsymbol{b}$ equals the expectations of the vector projected onto Gaussian random vectors. Our random slicing technique, which can be seen as a more general approach, has been inspired by the idea of Sliced Wasserstein distance(Rabin et al., 2012).

After that, the dot product of the Jacobian matrix and the random vector can be efficiently calculated with the assistance of lemma 3.3.

**Lemma 3.3** (Differential of Coordinate Transformation Function). $\forall \boldsymbol{x}, \boldsymbol{v} \in \mathbb{R}^{3N}$ *are a molecular conformation and Cartesian coordinate noise respectively, then*

$$J(\boldsymbol{x}) \cdot \boldsymbol{v} = f^{\mathcal{M}}(\boldsymbol{x} + \boldsymbol{v}) - f^{\mathcal{M}}(\boldsymbol{x}) + \alpha(\boldsymbol{x}; \boldsymbol{v}), \tag{16}$$

*where* $\alpha(\boldsymbol{x}; \boldsymbol{v}) = o(\boldsymbol{v})$ *as* $||\boldsymbol{v}|| \to 0$.

Therefore, the ultimate loss function can be defined as follows.

$$\mathcal{L}_{SliDe}(\mathcal{M}) = E_{p(\boldsymbol{x}|\boldsymbol{x}_0)} \frac{1}{N_v} \sum_{i=1}^{N_v} \left[ GNN_\theta(\boldsymbol{x})^\top \cdot \boldsymbol{v}_i - \frac{1}{\sigma} \nabla_{\boldsymbol{d}} E_{BAT}(\boldsymbol{d})^\top \cdot \left( f^{\mathcal{M}}(\boldsymbol{x} + \sigma \boldsymbol{v}_i) - f^{\mathcal{M}}(\boldsymbol{x}) \right) \right]^2, \tag{17}$$

where $\boldsymbol{v}_i \sim \mathcal{N}(\boldsymbol{0}, I_{3N})$, $\sigma$ is a parameter. $\boldsymbol{x}$ and $\boldsymbol{d}$ are the Cartesian coordinates and relative coordinates of the structure after adding the BAT noise to the equilibrium structure, respectively. $\nabla_{\boldsymbol{d}} E_{BAT}(\boldsymbol{d})$ is given in equation 10, and $GNN_\theta(\boldsymbol{x}) \in \mathbb{R}^{3N}$ denotes the prediction output of GNN for each atomic Cartesian coordinate.

The computation of the scalar target can be performed rapidly by leveraging the relative coordinate obtained after adding the Cartesian noise $v$ through the utilization of RDKit (Landrum et al., 2013), a readily available cheminformatics tool.

Consequently, the total loss is averaged on every sample in the 3D equilibrium molecular dataset $\mathbb{M}$:

$$\mathcal{L}_{SliDe}^{total} = \frac{1}{|\mathbb{M}|} \sum_{\mathcal{M} \in \mathbb{M}} \mathcal{L}_{SliDe}(\mathcal{M}). \tag{18}$$

In practise, we utilize stochastic gradient descent and approximate the total loss by batch loss. For reference, the pseudo-code outlining the approach for performing SliDe denoising pre-training is presented in Appendix C.2.

Furthermore, we have proven its equivalence to learning the force field of $E_{BAT}$ as shown in the following theorem, with the proof provided in appendix A.

**Theorem 3.4** (Interpretation of Sliced Denoising). *Given equilibrium structures, when* $\sigma$ *approaches* 0 *and* $N_{\boldsymbol{v}}$ *approaches* $\infty$, *minimizing* $\mathcal{L}_{SliDe}(\mathcal{M})$ *is equivalent to learning the force field of* $E_{BAT}$ *in Cartesian coordinate in equation 11.*

### 3.4 NETWORK ARCHITECTURE

Compared to previous denoising methods, our approach defines energy and noise w.r.t. relative coordinates. Relative coordinates provide a complete representation of molecular structure and conform the molecular symmetry, thereby offering advantages for molecular modeling. Further details about related work on 3D molecular modeling in relative coordinates can be found in Appendix D.2.

While TorchMD-NET (Thölke & Fabritiis, 2022) has achieved competitive results when applied in denoising tasks, as shown in (Zaidi et al., 2023; Feng et al., 2023), it employs Cartesian coordinates to inject geometry information and does not explicitly model the angles and torsion angles. Since our method explicitly utilizes relative coordinates to model energy and noise, we believe angular information is important for learning our force field target. Therefore, in addition to the vertex update in TorchMD-NET, we also incorporate edge update and introduce angular information in the edge embeddings. These edge embeddings are then utilized in the attention layer, which impacts the vertex update. Our network is denoted as the Geometric Equivariant Transformer (GET), and further details are outlined in Appendix C.3.

## 4 EXPERIMENTS

Our first experiment in section 4.1 is concerned with whether our approach achieves better physical consistency, specifically in terms of force field accuracy, as compared to coordinate denoising and fractional denoising methods. Then in section 4.2, we evaluate the performance of SliDe in comparison to state-of-the-art 3D pre-training methods on the benchmark datasets QM9 and MD17, in order to assess our model's ability for molecular property prediction. Furthermore, in section 4.3, we conduct ablation studies concerning fine-tuning regularization and network architecture. Additional experiments on the benchmark datasets ANI-1x, the impact of physical consistency on downstream tasks, and ablation studies related to the hyperparameters in SliDe loss, robustness to noisy pre-training data and effect of pre-training data scale can be found in the Appendix B. Implementation details of the experiments can be found in the Appendix C.

### 4.1 EVALUATIONS ON PHYSICAL CONSISTENCY

To estimate the learned force field in SliDe, we calculate the Cartesian force field for each molecule $\mathcal{M}$ by solving a least square estimation problem $\boldsymbol{A}\boldsymbol{x}_f = \boldsymbol{b}$, where $\boldsymbol{A} = [v_1, \cdots, v_{N_v}]^\top \in \mathbb{R}^{N_v \times 3N}$, $b_i = \frac{1}{\sigma}\nabla_{\boldsymbol{d}}E_{BAT}(\boldsymbol{d})^\top \left(f^{\mathcal{M}}(\boldsymbol{x} + \sigma\boldsymbol{v}_i) - f^{\mathcal{M}}(\boldsymbol{x})\right), \forall i = 1 \cdots N_v, \boldsymbol{b} = [b_1, \cdots, b_{N_v}]^\top \in \mathbb{R}^{N_v \times 3N}$. We can prove that the regression loss $E_{p(\mathcal{M}|\mathcal{M})}\left[GNN_\theta(\mathcal{M}) - \boldsymbol{x}_f\right]^2$ is asymptotically an equivalent optimization problem to SliDe. Therefore $\boldsymbol{x}_f$ can be viewed as the learned force field target in SliDe. Details can be found in appendix proposition A.1.

To verify the accuracy of the learned force field in various denoising methods, we compare the correlation coefficient between the learned force field and the ground-truth force field calculated by density functional theory (DFT). Since obtaining the true force field label can be time-consuming, the experiment is carried out on 1000 molecules randomly selected from dataset PCQM4Mv2 (Nakata & Shimazaki, 2017). The noisy conformations are generated according to each denoising method and the learned force fields of Frad and Coord are estimated as the approach in Feng et al. (2023).

Table 1: Correlation coefficient between the learned force field and the ground-truth force field of the three methods. The standard deviation is shown in parentheses. The top results are in bold.

| Denoising method | Coord | Frad | SliDe |
|---|---|---|---|
| Correlation coefficient | 0.616(0.047) | 0.631 (0.046) | **0.895** (0.071) |

The experimental results in Table 1 indicate that SliDe increases the correlation coefficient of the estimated force field by 42%, compared to Frad and Coord. This confirms that the design of our energy function and sliced denoising can help the model learn a more accurate force field than other denoising methods, which is consistent with our theoretical analysis. In addition, our result on Frad and Coord is in line with the experimental results in Feng et al. (2023), although the experiments are carried out on different datasets. It has been verified in Feng et al. (2023) that learning an accurate force field in denoising can improve downstream tasks. We also conduct a supplementary experiment in Appendix B.5 to confirm the conclusion. As a result, SliDe greatly improves the physical consistency of the denoising method, enabling the learned representations to have better performance on downstream tasks.

The validation of force field accuracy can also help us choose hyperparameters without training neural networks. Details are in Appendix B.2.

### 4.2 EVALUATIONS ON DOWNSTREAM TASKS

Our model is pre-trained on PCQM4Mv2 dataset (Nakata & Shimazaki, 2017), which contains 3.4 million organic molecules and provides one equilibrium conformation for each molecule. Following previous denoising methods, we apply the widely-used Noisy Nodes technique (Godwin et al., 2021), which incorporates coordinate denoising as an auxiliary task in addition to the original property prediction objective in the fine-tuning phase. Nevertheless, we observe the hard optimization of Noisy Nodes in SliDe. To get the most out of the fine-tuning technique, we add a regularization term in pre-training loss, i.e. $[GNN_\theta(\boldsymbol{x} + \tau\boldsymbol{v}) - \tau\boldsymbol{v}]^2$. An ablation study on the regularization term is

Table 2: Performance (MAE ↓) on QM9. The best results are in bold.

| | $\mu$ (D) | $\alpha$ ($a_0^3$) | homo (meV) | lumo (meV) | gap (meV) | $R^2$ ($a_0^2$) | ZPVE (meV) | $U_0$ (meV) | $U$ (meV) | $H$ (meV) | $G$ (meV) | $C_v$ ($\frac{cal}{mol \cdot K}$) |
|---|---|---|---|---|---|---|---|---|---|---|---|---|
| SchNet | 0.033 | 0.235 | 41.0 | 34.0 | 63.0 | 0.07 | 1.70 | 14.00 | 19.00 | 14.00 | 14.00 | 0.033 |
| E(n)-GNN | 0.029 | 0.071 | 29.0 | 25.0 | 48.0 | 0.11 | 1.55 | 11.00 | 12.00 | 12.00 | 12.00 | 0.031 |
| DimeNet++ | 0.030 | 0.044 | 24.6 | 19.5 | 32.6 | 0.33 | 1.21 | 6.32 | 6.28 | 6.53 | 7.56 | 0.023 |
| PaiNN | 0.012 | 0.045 | 27.6 | 20.4 | 45.7 | 0.07 | 1.28 | 5.85 | 5.83 | 5.98 | 7.35 | 0.024 |
| SphereNet | 0.025 | 0.045 | 22.8 | 18.9 | 31.1 | 0.27 | **1.120** | 6.26 | 6.36 | 6.33 | 7.78 | 0.022 |
| ET | 0.011 | 0.059 | 20.3 | 17.5 | 36.1 | **0.033** | 1.840 | 6.15 | 6.38 | 6.16 | 7.62 | 0.026 |
| TM | 0.037 | 0.041 | 17.5 | 16.2 | 27.4 | 0.075 | 1.18 | 9.37 | 9.41 | 9.39 | 9.63 | 0.022 |
| SE(3)-DDM | 0.015 | 0.046 | 23.5 | 19.5 | 40.2 | 0.122 | 1.31 | 6.92 | 6.99 | 7.09 | 7.65 | 0.024 |
| 3D-EMGP | 0.020 | 0.057 | 21.3 | 18.2 | 37.1 | 0.092 | 1.38 | 8.60 | 8.60 | 8.70 | 9.30 | 0.026 |
| Coord | 0.012 | 0.0517 | 17.7 | 14.3 | 31.8 | 0.4496 | 1.71 | 6.57 | 6.11 | 6.45 | 6.91 | 0.020 |
| Frad | 0.010 | 0.0374 | 15.3 | 13.7 | 27.8 | 0.3419 | 1.418 | 5.33 | 5.62 | 5.55 | 6.19 | 0.020 |
| SliDe | **0.0087** | **0.0366** | **13.6** | **12.3** | **26.2** | 0.3405 | 1.521 | **4.28** | **4.29** | **4.26** | **5.37** | **0.019** |

provided in section 4.3.1. Hyperparameter settings for pre-training and finetuning are summarized in Appendix C.4.

Our baselines include both 3D pre-training approaches, such as fractional denoising (Frad), coordinate denoising (Coord), 3D-EMGP (Jiao et al., 2023), SE(3)-DDM (Liu et al., 2023b), Transformer-M (TM) (Luo et al., 2023), as well as supervised models such as TorchMD-NET (ET) (Thölke & Fabritiis, 2022), SphereNet (Liu et al., 2022), PaiNN (Schütt et al., 2021), E(n)-GNN(Satorras et al., 2021), DimeNet (Gasteiger et al., 2020b), DimeNet++ (Gasteiger et al., 2020a), SchNet (Schütt et al., 2018). The results for these baselines are directly taken from the referred papers, except for Coord on MD17, which is produced by us due to its absence in their paper.

### 4.2.1 QM9

QM9 (Ramakrishnan et al., 2014) is a quantum chemistry dataset providing one equilibrium conformation and 12 labels of geometric, energetic, electronic, and thermodynamic properties for 134k stable small organic molecules made up of CHONF atoms. The data splitting follows standard settings which have a training set with 110,000 samples, a validation set with 10,000 samples, and a test set with the remaining 10,831 samples. The performance on 12 properties is measured by mean absolute error (MAE, lower is better) and the results are summarized in Table 2.

First of all, our model achieves new state-of-the-art performance on 10 out of 12 tasks in QM9, reducing the mean absolute error (MAE) by 12.4% compared to the existing state-of-the-art. Among them, SliDe performs particularly well on challenging energetic and thermodynamic tasks. We speculate that this is because these two tasks are more closely related to molecular potential energy and force fields that we focus on during pre-training, for instance, the potential energy is related to thermodynamic quantities as illustrated in (Saggion et al., 2019). It is worth noting that the downstream performance of the three interpretable methods, SliDe, Frad, and Coord, is in agreement with the result of learned force field accuracy in section 4.1, i.e. SliDe demonstrates the strongest performance while Frad outperforms Coord. These experimental findings once again confirm the importance of physical consistency to molecular representations.

### 4.2.2 MD17

MD17 (Chmiela et al., 2017) is a dataset of molecular dynamics trajectories of 8 small organic molecules. For each molecule, 150k to nearly 1M conformations, corresponding total energy and force labels are provided. We choose the challenging force prediction as our downstream task. The data splitting follows a standard limited data setting, where the model is trained on only 1000 samples, from which 50 are used for validation and the remaining data is used for testing. The performance is also measured by mean absolute error and the results are summarized in Table 3.

Despite the fact that the downstream task is closely related to our pre-training task, the input conformations in MD17 are far from equilibrium and the limited training data setting makes it even more challenging. In this case, we still outperform or achieve comparable results as compared with recent baselines, indicating that the force field knowledge learned in SliDe pre-training is effectively transferred to the downstream force field task.

Table 3: Performance (MAE ↓) on MD17 force prediction (kcal/mol/ Å). The best results are in bold. *: PaiNN does not provide the result for Benzene, and SE(3)-DDM utilizes the dataset for Benzene from Chmiela et al. (2018), which is a different version from ours (Chmiela et al., 2017).

|  | Aspirin | Benzene | Ethanol | Malonal -dehyde | Naphtha -lene | Salicy -lic Acid | Toluene | Uracil |
|---|---|---|---|---|---|---|---|---|
| SphereNet | 0.430 | 0.178 | 0.208 | 0.340 | 0.178 | 0.360 | 0.155 | 0.267 |
| SchNet | 1.35 | 0.31 | 0.39 | 0.66 | 0.58 | 0.85 | 0.57 | 0.56 |
| DimeNet | 0.499 | 0.187 | 0.230 | 0.383 | 0.215 | 0.374 | 0.216 | 0.301 |
| PaiNN* | 0.338 | - | 0.224 | 0.319 | 0.077 | 0.195 | 0.094 | 0.139 |
| ET | 0.2450 | 0.2187 | 0.1067 | 0.1667 | 0.0593 | 0.1284 | 0.0644 | 0.0887 |
| SE(3)-DDM* | 0.453 | - | 0.166 | 0.288 | 0.129 | 0.266 | 0.122 | 0.183 |
| Coord | 0.2108 | 0.1692 | 0.0959 | **0.1392** | 0.0529 | 0.1087 | 0.0582 | **0.0742** |
| Frad | 0.2087 | 0.1994 | 0.0910 | 0.1415 | 0.0530 | 0.1081 | **0.0540** | 0.0760 |
| SliDe | **0.1740** | **0.1691** | **0.0882** | 0.1538 | **0.0483** | **0.1006** | **0.0540** | 0.0825 |

## 4.3 ABLATION STUDY

We conduct an ablation study to examine the impact of the regularization term introduced for better fine-tuning and to evaluate the performance of our modified network architectures.

### 4.3.1 REGULARIZATION TERM

To assess the effectiveness of the regularization term proposed for pre-training SliDe, we conduct pre-training with and without regularization and subsequently fine-tuned the models on three QM9 tasks. The network architecture remains consistent across all three setups, and the Noisy Nodes are implemented with the same configuration. The result is shown in the bottom three rows of Table 4. Our findings indicate that the regularization term can effectively improve the performance of downstream tasks. Notably, SliDe without regularization still outperforms training from scratch and yields similar performance to Frad. Moreover, we observe in experiment that the regularization reduces the downstream Noisy Nodes loss, suggesting that the regularization term contributes to optimizing Noisy Nodes.

Table 4: Ablation study for regularization term.

| QM9 | homo | lumo | gap |
|---|---|---|---|
| Train from scratch | 17.6 | 16.7 | 31.3 |
| SliDe w/o regularization | 15.0 | 14.8 | 27.7 |
| SliDe w/ regularization | **13.6** | **12.3** | **26.2** |

Table 5: Ablation study for network design.

| MD17 force prediction | Aspirin | Benzene |
|---|---|---|
| SliDe (ET) | 0.2045 | 0.1810 |
| SliDe (GET) | **0.1740** | **0.1691** |

### 4.3.2 NETWORK DESIGN

To show the advantage of the improved network to our SliDe, we pre-train the geometric equivariant Transformer (GET) and TorchMD-NET (ET) by sliced denoising and fine-tune them on MD17. As shown in Table 5, our network further improves the performance, indicating the excellence of our novel network in depicting more intricate geometric features, such as angles and torsional angles.

## 5 CONCLUSION

This paper proposes a novel pre-training method, called sliced denoising, for molecular representation learning. Theoretically, it harbors a solid physical interpretation of learning force fields on molecular samples. The sampling distribution and regression targets are derived from classical mechanical molecular potential, ensuring more realistic input conformations and precise force field estimation than other denoising methods. Empirically, SliDe has shown significant improvements in force field estimation accuracy and various downstream tasks, including QM9 and MD17, as compared with previous supervised learning and pre-training methods.

ACKNOWLEDGEMENTS

This work is supported by National Key R&D Program of China No.2021YFF1201600 and Beijing Academy of Artificial Intelligence (BAAI).

We thank anonymous reviewers for constructive and helpful discussions.

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

# Appendix

## Table of Contents

# A    PROOF OF THEORETICAL RESULTS

*Proof of Lemma 3.3.* Since $f^{\mathcal{M}}$ is differentiable at the point $\boldsymbol{x}$, equation 16 is the definition of the differential of a function of several variables(Zorich, 2015). $\qquad\square$

*Proof of Lemma 3.2.*

$$E_{\boldsymbol{v}}[(\boldsymbol{a}-\boldsymbol{b})^{\top}\cdot\boldsymbol{v}]^2 = E_v[(\boldsymbol{a}-\boldsymbol{b})^{\top}\boldsymbol{v}\boldsymbol{v}^{\top}(\boldsymbol{a}-\boldsymbol{b})] = (\boldsymbol{a}-\boldsymbol{b})^{\top}E_v[\boldsymbol{v}\boldsymbol{v}^{\top}](\boldsymbol{a}-\boldsymbol{b})$$
$$= (\boldsymbol{a}-\boldsymbol{b})^{\top}\sigma^2 I_{3N}(\boldsymbol{a}-\boldsymbol{b}) = \sigma^2||\boldsymbol{a}-\boldsymbol{b}||^2 \tag{19}$$

Divide both sides by $\sigma^2$, then the proof is completed. $\qquad\square$

*Proof of Theorem 3.4.* Let $\boldsymbol{v}^{(\sigma)} \triangleq \sigma\boldsymbol{v}$, $\boldsymbol{v} \sim N(0, I_{3N})$.

$$\mathcal{L}_{SliDe}(\mathcal{M}) \approx E_{p(\boldsymbol{x}|\boldsymbol{x}_0)}E_{\boldsymbol{v}}\left[GNN_\theta(\boldsymbol{x})^{\top}\cdot\boldsymbol{v} - \frac{1}{\sigma}\nabla_{\boldsymbol{d}}E_{BAT}(\boldsymbol{d})^{\top}\cdot\left(f^{\mathcal{M}}(\boldsymbol{x}+\sigma\boldsymbol{v}) - f^{\mathcal{M}}(\boldsymbol{x})\right)\right]^2 \tag{20}$$

$$= E_{p(\boldsymbol{x}|\boldsymbol{x}_0)}E_{\boldsymbol{v}^{(\sigma)}}\frac{1}{\sigma^2}\left[GNN_\theta(\boldsymbol{x})^{\top}\cdot\boldsymbol{v}^{(\sigma)} - \nabla_{\boldsymbol{d}}E_{BAT}(\boldsymbol{d})^{\top}\cdot\left(f^{\mathcal{M}}(\boldsymbol{x}+\boldsymbol{v}^{(\sigma)}) - f^{\mathcal{M}}(\boldsymbol{x})\right)\right]^2 \tag{21}$$

$$\approx E_{p(\boldsymbol{x}|\boldsymbol{x}_0)}E_{\boldsymbol{v}^{(\sigma)}}\frac{1}{\sigma^2}\left[GNN_\theta(\boldsymbol{x})^{\top}\cdot\boldsymbol{v}^{(\sigma)} - \nabla_{\boldsymbol{d}}E_{BAT}(\boldsymbol{d})^{\top}\cdot\left(J(\boldsymbol{x})\boldsymbol{v}^{(\sigma)}\right)\right]^2 \tag{22}$$

$$= E_{p(\boldsymbol{x}|\boldsymbol{x}_0)}E_{\boldsymbol{v}^{(\sigma)}}\frac{1}{\sigma^2}\left[\left(GNN_\theta(\boldsymbol{x}) - J(\boldsymbol{x})^{\top}\cdot\nabla_{\boldsymbol{d}}E_{BAT}(\boldsymbol{d})\right)^{\top}\cdot\boldsymbol{v}^{(\sigma)}\right]^2 \tag{23}$$

$$= E_{p(\boldsymbol{x}|\boldsymbol{x}_0)}||GNN_\theta(\boldsymbol{x}) - \nabla_{\boldsymbol{x}}E_{BAT}(\boldsymbol{d}(\boldsymbol{x}))||^2 \tag{24}$$

The first step of approximation holds in the sense that $\lim_{N_v\to\infty}\mathcal{L}_{SliDe} = equation\ 20$. The second step holds by substituting $\sigma\boldsymbol{v}$ with $\boldsymbol{v}^{(\sigma)} \sim N(0, \sigma^2 I_{3N})$. The third step holds because of Lemma 3.3 and approximation holds in the sense that $\lim_{\sigma\to0}\alpha(\boldsymbol{x};\boldsymbol{v}^{(\sigma)}) \to 0$. The fourth step of the equation is the associative and distributive law of vector multiplication. The last step uses Lemma 3.2. $\qquad\square$

**Proposition A.1.** *When $N_v \to \infty$ and the least square estimation referred in section 4.1 gives the ground-truth force field, i.e. $\boldsymbol{A}\boldsymbol{x}_f = \boldsymbol{b}$ for any sampled $\boldsymbol{v}$, the following regression loss is an equivalent optimization problem to SliDe.*

$$\mathcal{L}_{SliDe}^{(reg)}(\mathcal{M}) = E_{p(\mathcal{M}|\mathcal{M})}\left[GNN_\theta(\mathcal{M}) - \boldsymbol{x}_f\right]^2, \tag{25}$$

*Proof.* When $N_v \to \infty$, by the law of large numbers,

$$\lim_{N_v\to\infty}\mathcal{L}_{SliDe}(\mathcal{M}) = \lim_{N_v\to\infty}E_{p(\boldsymbol{x}|\boldsymbol{x}_0)}\frac{1}{N_v}\sum_{i=1}^{N_v}[\boldsymbol{A}\cdot GNN_\theta(\boldsymbol{x}) - \boldsymbol{b}]^2 \tag{26}$$

$$= E_{p(\boldsymbol{x}|\boldsymbol{x}_0)}E_{\boldsymbol{v}}[\boldsymbol{A}\cdot GNN_\theta(\boldsymbol{x}) - \boldsymbol{b}]^2 \triangleq \mathcal{L}_{SliDe}^{(asymp)}(\mathcal{M}) \tag{27}$$

$$\nabla\mathcal{L}_{SliDe}^{(reg)}(\mathcal{M}) = E_{p(\boldsymbol{x}|\boldsymbol{x}_0)}2\left(GNN_\theta(\boldsymbol{x}) - \boldsymbol{x}_f\right)^{\top}\nabla GNN_\theta(\boldsymbol{x}) \tag{28}$$

$$\nabla\mathcal{L}_{SliDe}^{(asymp)}(\mathcal{M}) = E_{p(\boldsymbol{x}|\boldsymbol{x}_0)}E_{\boldsymbol{v}}2\left(\boldsymbol{A}\cdot GNN_\theta(\boldsymbol{x}) - \boldsymbol{b}\right)^{\top}\boldsymbol{A}\nabla GNN_\theta(\boldsymbol{x}) \tag{29}$$

By assumption, $\boldsymbol{A}\boldsymbol{x}_f = \boldsymbol{b}$ for any sampled $\boldsymbol{v}$, then equation 29

$$= E_{p(\boldsymbol{x}|\boldsymbol{x}_0)}E_{\boldsymbol{v}}2\left(GNN_\theta(\boldsymbol{x}) - \boldsymbol{x}_f\right)^{\top}\boldsymbol{A}^{\top}\boldsymbol{A}\nabla GNN_\theta(\boldsymbol{x}) \tag{30}$$

$$= E_{p(\boldsymbol{x}|\boldsymbol{x}_0)}2\left(GNN_\theta(\boldsymbol{x}) - \boldsymbol{x}_f\right)^{\top}E_{\boldsymbol{v}}\left[\boldsymbol{A}^{\top}\boldsymbol{A}\right]\nabla GNN_\theta(\boldsymbol{x}) \tag{31}$$

Since $\boldsymbol{v}_i \sim N(0, I_{3N})$, every element in $\boldsymbol{A} = [v_1, \cdots, v_{N_v}]$ is i.i.d standard normal distribution. Therefore $E_{\boldsymbol{v}}\left[\boldsymbol{A}^{\top}\boldsymbol{A}\right] = N_v \cdot \boldsymbol{I}_{3N}$, i.e. equation 31

$$= E_{p(\boldsymbol{x}|\boldsymbol{x}_0)}2N_v\left(GNN_\theta(\boldsymbol{x}) - \boldsymbol{x}_f\right)^{\top}\nabla GNN_\theta(\boldsymbol{x}) \tag{32}$$

$$= N_v\nabla\mathcal{L}_{SliDe}^{(reg)}(\mathcal{M}) \tag{33}$$

Consequently, $\nabla\mathcal{L}_{SliDe}^{(asymp)} = N_v \nabla\mathcal{L}_{SliDe}^{(reg)}$, $N_v$ is a constant, then the two optimization target share the same minima. □

**Theorem A.2** (Interpretation of Coordinate Denoising (Zaidi et al., 2023)). *Assume the conformation distribution is a mixture of Gaussian distribution centered at the equilibriums:*

$$p(\boldsymbol{x}) = \int p(\boldsymbol{x}|\boldsymbol{x}_0)p(\boldsymbol{x}_0), \; p(\boldsymbol{x}|\boldsymbol{x}_0) \sim \mathcal{N}(\boldsymbol{x}_0, \tau_c^2 I_{3N}) \tag{34}$$

$\boldsymbol{x}_0, \; \boldsymbol{x} \in \mathbb{R}^{3N}$ *are equilibrium conformations and noisy conformation respectively, $N$ is the number of atoms in the molecule. It relates to molecular energy by Boltzmann distribution $p(\boldsymbol{x}) \propto exp(-E_{Coord}(\boldsymbol{x}))$.*

*Then given a sampled molecule $\mathcal{M}$, the coordinate denoising loss is an equivalent optimization target to force field regression:*

$$\mathcal{L}_{Coord}(\mathcal{M}) = E_{p(\boldsymbol{x}|\boldsymbol{x}_0)p(\boldsymbol{x}_0)}||GNN_\theta(\boldsymbol{x}) - (\boldsymbol{x} - \boldsymbol{x}_0)||^2 \tag{35}$$

$$\simeq E_{p(\boldsymbol{x})}||GNN_\theta(\boldsymbol{x}) - (-\nabla_{\boldsymbol{x}}E_{Coord}(\boldsymbol{x}))||^2, \tag{36}$$

*where $GNN_\theta(\boldsymbol{x})$ denotes a graph neural network with parameters $\theta$ which takes conformation $\boldsymbol{x}$ as an input and returns node-level noise predictions, $\simeq$ denotes equivalent optimization objectives for GNN.*

*Proof.* According to Boltzmann distribution, equation 36= $E_{p(\boldsymbol{x})}||GNN_\theta(\boldsymbol{x}) - \nabla_{\boldsymbol{x}}\log p(\boldsymbol{x})||^2$. By using a conditional score matching lemma (Vincent, 2011), the equation above = $E_{p(\boldsymbol{x}|\boldsymbol{x}_0)p(\boldsymbol{x}_0)}||GNN_\theta(\boldsymbol{x}) - \nabla_{\boldsymbol{x}}\log p(\boldsymbol{x}|\boldsymbol{x}_0)||^2 + T_1$, where $T_1$ is constant independent of $\theta$. Then with the Gaussian assumption, it becomes $E_{p(\boldsymbol{x}|\boldsymbol{x}_0)p(\boldsymbol{x}_0)}||GNN_\theta(\boldsymbol{x}) - \frac{\boldsymbol{x}_0 - \boldsymbol{x}}{\tau_c^2}||^2 + T_1$. Finally, since coefficients $-\frac{1}{\tau^2}$ do not rely on the input $\boldsymbol{x}$, it can be absorbed into $GNN_\theta$, thus obtaining equation 35. □

**Theorem A.3** (Interpretation of Fractional Denoising (Feng et al., 2023)). *Assume the conformation distribution is a mixture distribution centered at the equilibriums:*

$$p(\boldsymbol{x}) = \int p(\boldsymbol{x}|\boldsymbol{x}_a)p(\boldsymbol{x}_a|\boldsymbol{x}_0)p(\boldsymbol{x}_0), \; p(\boldsymbol{\psi}_a|\boldsymbol{\psi}_0) \sim \mathcal{N}(\boldsymbol{\psi}_0, \sigma_f^2 I_m), \; p(\boldsymbol{x}|\boldsymbol{x}_a) \sim \mathcal{N}(\boldsymbol{x}_a, \tau_f^2 I_{3N}), \tag{37}$$

*where $\boldsymbol{x}_0, \; \boldsymbol{x}_a, \; \boldsymbol{x} \in \mathbb{R}^{3N}$ are equilibrium conformation and noisy conformations respectively, $\boldsymbol{\psi}$ and $\boldsymbol{\psi}_0$ are the dihedral angles of rotatable bonds in conformation $\boldsymbol{x}$ and $\boldsymbol{x}_0$, $m$ is the number of the rotatable bonds. It relates to molecular energy by Boltzmann distribution $p(\boldsymbol{x}) \propto exp(-E_{Frad}(\boldsymbol{x}))$.*

*Then given a sampled molecule $\mathcal{M}$, the fractional denoising loss is an equivalent optimization target to force field regression:*

$$\mathcal{L}_{Frad}(\mathcal{M}) = E_{p(\boldsymbol{x}|\boldsymbol{x}_a)p(\boldsymbol{x}_a|\boldsymbol{x}_0)p(\boldsymbol{x}_0)}||GNN_\theta(\boldsymbol{x}) - (\boldsymbol{x} - \boldsymbol{x}_a)||^2 \tag{38}$$

$$\simeq E_{p(\boldsymbol{x})}||GNN_\theta(\boldsymbol{x}) - (-\nabla_{\boldsymbol{x}}E_{Frad}(\boldsymbol{x}))||^2, \tag{39}$$

*Proof.* According to Boltzmann distribution, equation 39= $E_{p(\boldsymbol{x})}||GNN_\theta(\boldsymbol{x}) - \nabla_{\boldsymbol{x}}\log p(\boldsymbol{x})||^2$. By using a conditional score matching lemma (Vincent, 2011), the equation above = $E_{p(\boldsymbol{x},\boldsymbol{x}_a)}||GNN_\theta(\boldsymbol{x}) - \nabla_{\boldsymbol{x}}\log p(\boldsymbol{x}|\boldsymbol{x}_a)||^2 + T_2$, where $T_2$ is constant independent of $\theta$. Since the part in the expectation does not contain $\boldsymbol{x}_0$, it equals to $E_{p(\boldsymbol{x},\boldsymbol{x}_a,\boldsymbol{x}_0)}||GNN_\theta(\boldsymbol{x}) - \nabla_{\boldsymbol{x}}\log p(\boldsymbol{x}|\boldsymbol{x}_a)||^2 + T_2$. Finally, $-\frac{1}{\tau^2}$ can be absorbed into GNN and obtain equation 38. □

# B  SUPPLEMENTARY EXPERIMENTS

## B.1  MORE DOWNSTREAM RESULTS

We evaluate SliDe on ANI-1x that provides large numbers of molecules with multiple equilibrium and nonequilibrium conformations. A summary of datasets is shown in table 6. The result is compared with baseline nonequilibrium denoising (Noneq) Wang et al. (2023b) that performs coordinate denoising as Coord Zaidi et al. (2023), while Noneq is pre-trained on ANI-1 and ANI-1x that

Table 6: Summary of datasets.

| Dataset | Molecule size | Number of molecules | Number of conformations |
|---|---|---|---|
| PCQM4Mv2 | $\sim 30$ | 3,378,606 | 3,378,606(one equilibrium for each molecule) |
| QM9 | $\sim 18$ (3-29) | 133,885 | 133,885(one equilibrium for each molecule) |
| MD17 | $\sim 13$ (9-21) | 8 | 3,611,115(including nonequilibriums) |
| ANI-1x | $\sim 15$ (4-63) | 63,865 | 5,496,771(including nonequilibriums) |

contain nonequilibrium structures. Noneq also adopts TorchMD-NET as the backbone model. The result of experiment on ANI-1x energy prediction is shown in table 7. SliDe performs better in both pre-train improvement and without pre-train setting, indicating the superiority of SliDe pre-training method over nonequilibrim coordiante denoising as well as the improved backbone model GET over TorchMD-NET.

Table 7: Performance (MAE $\downarrow$) on ANI-1x energy prediction (kcal/mol). The best results are in bold.

| | Noneq | SliDe |
|---|---|---|
| w/o pre-train | 1.50 | 1.362 |
| pre-train | 1.01 | 0.786 |
| pre-train improvement | 32.7% | **42.3%** |

## B.2 HYPERPARAMETER ANALYSIS OF THE SLIDE LOSS FUNCTION

Table 8: Force field accuracy in different settings of $N_v$ and $\sigma$. The top results are in bold.

| Settings | $\rho$ | MSE | Scale |
|---|---|---|---|
| $N_v = 32, \sigma = 0.001$ | 0.536(0.067) | 2.3e-4(1e-4) | 0.73(0.13) |
| $N_v = 64, \sigma = 0.001$ | 0.753(0.079) | 1.5e-4(8e-5) | **0.98** (0.14) |
| $N_v = 128, \sigma = 0.001$ | 0.895 (0.071) | **7.5e-5**(2e-4) | 1.05(0.14) |
| $N_v = 512, \sigma = 0.001$ | **0.896** (0.067) | 7.6e-5(7e-5) | 1.06(0.15) |
| $N_v = 512, \sigma = 0.01$ | 0.893(0.072) | 0.53(0.20) | 41.97(5.70) |

Since an accurate force field target contributes to learning effective representations, we can choose hyperparameters by utilizing the least square estimation of learned force field accuracy introduced in section 4.1. This parameter selection strategy obviates the need for training neural networks, thereby rendering the process efficient and principled. Accordingly, we validated the accuracy of the learned force field for several combinations of hyperparameters $N_v$ and $\sigma$. The results are shown in Table 8. The accuracy is measured by Pearson correlation coefficient ($\rho$, the larger the better), mean squared error (MSE, the smaller the better), and "scale", which is the quotient of the mean absolute values between learned force fields and DFT force fields, and the best value is 1. The value in the bracket is the standard deviation.

In Theorem 3.4, the best hyperparameter in theory is $N_v \to \infty$ and $\sigma \to 0$. However, a large sampling size $N_v$ leads to slow pre-training, and a small sampling standard deviation $\sigma$ results in higher numerical accuracy required. In experimental results, larger $N_v$ leads to better force field accuracy but the trend tends to saturate when $N_v > 512$. This is mainly because in the pre-training dataset, the atomic numbers of the molecules are generally distributed between 20 and 40, i.e. $N_v = 128 > 3N$ for most molecules and the least square error is small in this case. As for the standard deviation $\sigma$, it has a small impact on the correlation coefficient, but significantly affects the MSE and scale. After considering both accuracy and efficiency, we choose $N_v = 128$ and $\sigma = 0.001$.

### B.3 ROBUSTNESS TO NOISY PRE-TRAINING DATA

In practical applications, the data quality can vary significantly. Most of the existing methods rely on equilibrium structures, limiting their effectiveness in scenarios where such structures are not readily available or accurate. To investigate the robustness of SliDe to noisy conformation, we pre-train on RDKit generated conformations, which can be generated at a large scale. We extract subsets of 10W and 50W molecules from PCQM4Mv2 dataset and generate cheap conformations by RDKit. We pre-train on the dataset we construct and test them on homo(QM9), as shown in table 9. When compared with training from scratch MAE=17.6 meV, pre-training with RDKit data

Table 9: Performance (MAE ↓) on homo (meV) from QM9 with SliDe pre-trained on accurate and noisy conformations.

| Training size | PCQ(DFT) | PCQ(RDKit) |
|---|---|---|
| 10W | 16.05 | 16.13 |
| 50W | 14.53 | 15.39 |

is notably effective. When compared with pre-training with accurate conformations, less accurate conformations compromise the performance. Overall, SliDe is robust to inaccurate conformations, revealing the potential of SliDe in larger scale of pre-training.

### B.4 EFFECT OF PRE-TRAINING DATA SCALE ON DOWNSTREAM TASKS

To investigate how the scale of the pre-training data affects performance, we randomly extract 10W, 50W, 100W data from PCQM4Mv2 dataset. We pre-train on these subsets and finetune on homo(QM9). The results is provided in table 10. We find more pre-train data contributes to better downstream performance, whereas SliDe is able to achieve competitive results when data is scarce, such as 50W.

Table 10: Performance (MAE ↓) on homo (meV) from QM9 with SliDe pre-trained on different data scales.

| Number of pre-training data | w/o pretrain | 10W | 50W | 100W | 300W(whole) |
|---|---|---|---|---|---|
| homo(meV) | 17.6 | 16.05 | 14.53 | 14.21 | 13.6 |

### B.5 EFFECT OF PRE-TRAINING PHYSICAL CONSISTENCY ON DOWNSTREAM TASKS

Table 11: Performance (MAE ↓) on MD17 force prediction (kcal/mol/ Å). The models are pre-trained on a subset of PCQM4Mv2 dataset.

| | Train from Scratch | Coord pre-training | Frad pre-training | DFT label supervised pre-training |
|---|---|---|---|---|
| Aspirin (Force) | 0.253 | 0.250 | 0.248 | 0.236 |

To verify whether learning an accurate force field in denoising can improve downstream tasks, we compare the existing denoising method with supervised pre-training on precise force field labels by DFT. Since DFT calculation can be time-consuming, we randomly select 10,000 molecules with fewer than 30 atoms from the PCQM4Mv2 dataset and calculate their precise force field label using DFT. We pre-train the model by learning the DFT force labels. We also pre-train the same backbone model by Coord and Frad methods respectively. Then the pre-trained models are finetuned on MD17 datasets. The results are shown in Table 11. The comparison between the pre-training methods indicates that as the accuracy of the force field in pre-training tasks increases, the performance on downstream tasks improves. Note that there is a large gap from Frad to "DFT label supervised"

compared to the improvement from training from Scratch to Coord and Coord to Frad, indicating that there is still a large room for downstream improvement along the idea of learning force fields. These findings motivate us to design a denoising pre-training task to learn accurate force fields.

## C  DETAIL IMPLEMENTATIONS

### C.1  NOISE DESIGN

The BAT noise strategy refers to perturbing the equilibrium structure by adding independent Gaussian noise on every bond length, angle and torsion angle, as shown in equation 9. The variances are determined by the parameters that can be obtained from the parameter files of molecular simulation tools. Our parameters are obtained from the parameter files of Open Force Field v.2.0.0 (Sage) (Boothroyd et al., 2023). Examples of BAT noise on acetaldehyde and 2-methylpyridine are provided in figure 2 to illustrate the noise scales.

In some cases, the independence is unable to be achieved. We make special treatments for these situations. Firstly, when an atom is connected to more than two atoms, the angles centered on this atom are dependent. In this case, we randomly select one edge and add noise to the angles involving this edge. For example, when adding angle noise on the methyl group "-$CH_3$" in figure 2b, the bond "C-C" is selected to be fixed and add noise on three angles of "C-C-H". Secondly, when there are ring structures in the molecule, the bond lengths, bond angles, and dihedral angles formed by the atoms in the ring are dependent. As a solution, we do not add noise to the bonds, angles, and torsion angles that are inside the ring. For example, in figure 2c, no noise is added on the pyridine backbone. One reason is that RDkit does not support modifying bond lengths, bond angles, and dihedral angles inside rings. The other reason is that we attempt to add low-level independent Gaussian Cartesian coordinate noise to the atoms inside the ring to perturb them sufficiently. However, we find that its force field accuracy on certain molecules is much lower than without adding noise inside the ring. We speculate that this is because perturbing the atomic coordinates in the ring affects the surrounding angles and torsion angles.

### C.2  PSEUDOCODE FOR ALGORITHMS AND COMPLEXITY ANALYSIS

In this section, we present pseudocode to illustrate the pre-training algorithm of SliDe in Algorithm 1. Besides, to show the ability to apply to large molecules, we discuss the time complexity of SliDe algorithm and provide a comparison between Frad and SliDe.

First of all, please note that all the terms in equation 17 except for $GNN_\theta(x)$, are calculated in data preprocessing. As the deep learning package (specifically PyTorch in this case) generates multiple workers in the DataLoader through parallel processes, each responsible for loading and processing individual data samples before adding the processed batches to a queue, this approach ensures that the time-consuming training process remains unimpeded, as long as the queue is consistently filled with batches. Therefore, it will not become the bottleneck of the training process, as $GNN_\theta(x)$ is what takes most of the time.

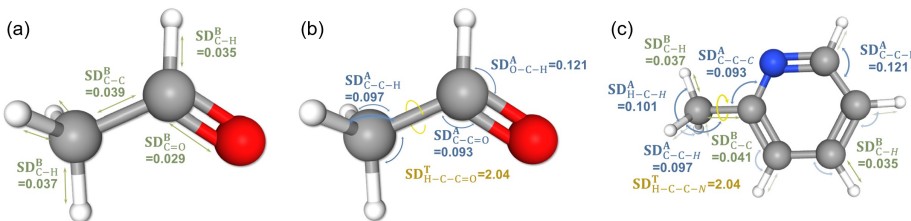

Figure 2: Some examples of BAT noise on molecules. The scale of noise can distinguish different bond types and atom types, resulting in better molecular modeling. (a)(b) Standard deviations of BAT noise for acetaldehyde. (c) Standard deviations of BAT noise for 2-methylpyridine.

---

**Algorithm 1** Sliced Denoising Pre-training Algorithm

---

**Require:**

    $GNN$: Graph Neural Network

    $\mathbb{M}$: Unlabeled 3D molecular pre-training dataset

    $T$: Training steps

    $BS$: Batch size

    $\sigma$: Standard deviation of the sampled coordinate noise

    $N_v$: Sample number of coordinate noise

    $\mathcal{N}$: Gaussian distribution

 1: **while** $T \neq 0$ **do**

 2:      Randomly sample a batch of molecules from the dataset $\mathbb{M}$.

 3:      **for** each molecule $\mathcal{M}$ with equilibrium structure $\boldsymbol{x}_0$ in the batch **do**

 4:          Get the bond lengths, angles, torsion angles of $\mathcal{M}$, denoted as $(\boldsymbol{r}_0, \boldsymbol{\theta}_0, \boldsymbol{\phi}_0)$.

 5:          Get the parameters of $\mathcal{M}$, denoted as $(\boldsymbol{k}^B, \boldsymbol{k}^A, \boldsymbol{k}^T, \boldsymbol{\omega})$.

 6:          Add BAT noise on the structure according to equation 9 and get the perturbed structure $\boldsymbol{x}$ and $\boldsymbol{d}$

 7:          Calculate $\nabla_{\boldsymbol{d}} E_{BAT}(\boldsymbol{d})$ according to equation 10.

 8:          **for** i = 1, ..., $N_v$ **do**

 9:              Sample coordinate noise $\boldsymbol{v}_i \sim \mathcal{N}(\boldsymbol{0}, I_{3N})$

10:              Calculate its corresponding relative coordinate changes $f^{\mathcal{M}}(\boldsymbol{x} + \sigma \boldsymbol{v}_i) - f^{\mathcal{M}}(\boldsymbol{x})$

11:              $Loss_i^{\mathcal{M}} = \left[ GNN_\theta(\boldsymbol{x})^\top \cdot \boldsymbol{v}_i - \frac{1}{\sigma} \nabla_{\boldsymbol{d}} E_{BAT}(\boldsymbol{d})^\top \cdot \left( f^{\mathcal{M}}(\boldsymbol{x} + \sigma \boldsymbol{v}_i) - f^{\mathcal{M}}(\boldsymbol{x}) \right) \right]^2 + \left[ GNN_\theta(\boldsymbol{x} + \tau \boldsymbol{v}_i) - \tau \boldsymbol{v}_i \right]^2$

12:          **end for**

13:      **end for**

14:      Optimise $Loss = \frac{1}{N_v \times BS} \sum_{\mathcal{M} \in batch} \sum_{i=1}^{N_v} Loss_i^{\mathcal{M}}$ and update GNN

15:      $T = T - 1$

16: **end while**

---

In theory, the computational complexity of the regression target for one molecule is $O(N_v \times \max\{N, M, F\})$ for SliDe, where N is the number of atoms, M is the dimension of relative coordinates (total number of bond lengths, bond angles, and bond torsions), F is the complexity of coordinate transformation between Cartesian coordinates and relative coordinates.

Two factors affecting the complexity is $N_v$ and $F$. As for coordinate transformation, the complexity $F = O(\max N, M_t)$ Choi (2006), which is executed efficiently by RDKit. As for the $N_v$ times sampling procedure, in fact, for every $i \in \{1, \cdots, N_v\}$, the regression target can be calculated in parallel. Therefore the time complexity above can be further reduced to $O(\max N, M, F)$. This makes pre-training on large molecular datasets possible.

As for comparison to existing method, SliDe and Frad are theoretically in the same scale. The computational complexity of the regression target for one molecule is $O(\max\{M_{rb}, N, F_{rb}\})$ for Frad, where $M_{rb}$ is the number of rotatable bonds and $F_{rb}$ is the coordinate transformation between Cartesian coordiantes and diheral angles of rotatable bonds. It is in the same scale to SliDe's $O(\max\{N, M, F\})$.

In practise, although we do not process all $N_v$ targets in parallel and incorporate edge update in network architecture, the training time of Frad and SliDe does not vary significantly. Frad pre-training takes 1d1h14m on 8 NVIDIA A100 GPU, and SliDe pre-training takes 1d17h1m on 8 Tesla V100 GPU.

## C.3  Architecture Details

Our network is an equivariant graph neural network that recursively updates vertex and edge features. An illustration of the network architecture is shown in Figure 3. The vertex feature $V \in \mathbb{R}^{3 \times F_V}$ and $S \in \mathbb{R}^{F_V}$ are respectively vector and scalar features for each vertex, $F_V$ is the vertex feature dimension. $E \in \mathbb{R}^{F_E}$ denotes the edge feature of each edge, $F_E$ is the edge feature dimension. The edge vector $\boldsymbol{x}_i - \boldsymbol{x}_j$ is denoted by $R$. $Z$ is the atomic type. $\boldsymbol{r}, \boldsymbol{\theta}, \boldsymbol{\phi}$ are bond lengths, bond angles and torsion angles.

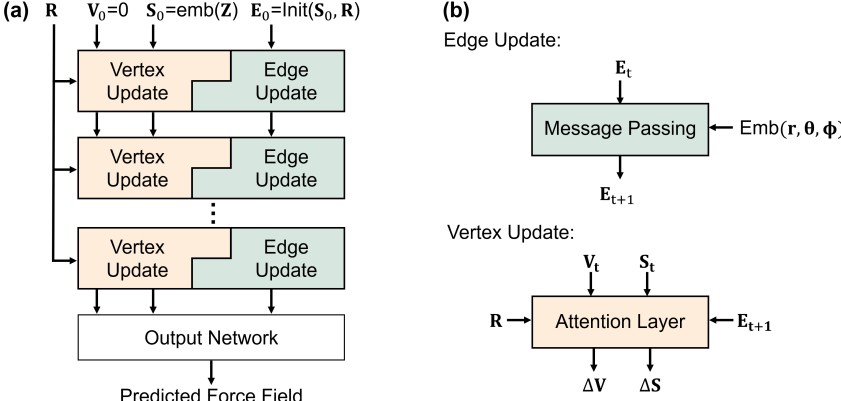

Figure 3: Overview of the network architecture. (a) The whole architecture includes initialization, several update layers and an output network. (b) The update layer consists of edge and vertex updates. The updated edge feature will be used in vertex updates.

For the edge update, the invariant edge feature $E$ is updated by the embeddings of the bond length $r$, the neighbor edge features and the embeddings of their angles $\theta, \phi$. Specifically, it incorporates Bessel functions to embed bond length and spherical harmonics to embed angle and torsion angles, which are shown to be effective geometry embeddings for molecular representation (Liu et al., 2022; Klicpera et al., 2021; Gasteiger et al., 2020b). For the vertex update, the invariant $S$ and equivariant $V$ are updated by an attention layer, whose architecture is based on TorchMD-NET. The updated edge features are projected into two filters and are later used to calculate attention weights in vertex update.

## C.4 HYPERPARAMETER SETTINGS

Table 12: Hyperparameters for pre-training.

| Parameter | Value or description |
|---|---|
| Train Dataset | PCQM4MV2 |
| Batch size | 128 |
| Optimizer | AdamW |
| Warm up steps | 10000 |
| Max Learning rate | 0.0004 |
| Learning rate decay policy | Cosine |
| Learning rate factor | 0.8 |
| Cosine cycle length | 240000 |
| Network structure | Keep aligned with downstream settings respectively on QM9 and MD17 |
| $N_v$ | 128 |
| $\sigma$ | 0.001 |
| Regression target | Least square results* |
| Regularization$^\dagger$ | yes |
| $\tau$ | 0.04 |

Hyperparameters for pre-training are listed in Table 12. Details about Learning rate decay policy can be refered in https://hasty.ai/docs/mp-wiki/scheduler/reducelronplateau#strong-reducelronplateau-explained-strong.

*: In previous denoising methods, normalizing the regression target, such as noise in Coord, is a widely applied technique to stabilize the training process. However, in SliDe loss equation 17, the

normalization is hard to implement. Instead, we chose to utilize $\mathcal{L}_{SliDe}^{(reg)}$ and normalize the regression target $\boldsymbol{x}_f$. We find the least squares estimation does not incur significant additional computational costs on the current dataset of molecular sizes.

[†]: In order to align with the downstream Noisy Node task that involves coordinate denoising, we add a regularization term in pre-training. So the total pre-training loss is given by

$$
\begin{aligned}
E_{p(\boldsymbol{x}|\boldsymbol{x}_0)} \frac{1}{N_v} \sum_{i=1}^{N_v} &\left\{ \left[ GNN_\theta(\boldsymbol{x} + \tau \boldsymbol{v}_i) - \tau \boldsymbol{v}_i \right]^2 \right. \\
&\left. + \left[ GNN_\theta(\boldsymbol{x})^\top \cdot \boldsymbol{v}_i - \frac{1}{\sigma} \nabla_{\boldsymbol{d}} E_{BAT}(\boldsymbol{d})^\top \cdot \left( f^{\mathcal{M}}(\boldsymbol{x} + \sigma \boldsymbol{v}_i) - f^{\mathcal{M}}(\boldsymbol{x}) \right) \right]^2 \right\}
\end{aligned}
\tag{40}
$$

Table 13: Hyperparameters for fine-tuning on MD17.

| Parameter | Value or description |
|---|---|
| Train/Val/Test Splitting | 950/50/remaining data |
| Batch size | 8 |
| Optimizer | AdamW |
| Warm up steps | 1000 |
| Max Learning rate | 0.0005 |
| Learning rate decay policy | ReduceLROnPlateau (Reduce Learning Rate on Plateau) scheduler |
| Learning rate factor | 0.8 |
| Patience | 30 |
| Min learning rate | 1.00E-07 |
| Network structure | Geometric Equivariant Transformer |
| Head number | 8 |
| Layer number | 6 |
| RBF number | 32 |
| Activation function | SiLU |
| Embedding dimension | 128 |
| Force weight | 0.8 |
| Energy weight | 0.2 |
| Noisy Nodes(NN) denoise weight | 0.1 |
| Dihedral angle noise scale in NN | 20 |
| Coordinate noise scale in NN | 0.005 |

Hyperparameters for fine-tuning on MD17 are listed in Table 13.

Hyperparameters for fine-tuning on QM9 are listed in Table 14. The cosine cycle length is set to be $500000$ for $\alpha$, $ZPVE$, $U_0$, $U$, $H$, $G$ and $300000$ for other tasks for fully converge. Following previous literature (Liu et al., 2023b; Feng et al., 2023; Liu et al., 2022), we fix seed= $1$ on QM9 and MD17, as the performance is quite stable for random seeds (Schütt et al., 2018; 2021; Liu et al., 2022). To further validate the issue, we finetune with seed= $0, 1, 2$ on homo(QM9), lumo(QM9) and Aspirin(MD17). The results is quite stable, as shown in Table 15

For experiment in section 4.1, the force field calculated by DFT method is implemented by PySCF tool (Sun et al., 2020), with basis = '6-31g', xc = 'b3lyp'.

Noisy Nodes is implemented following (Godwin et al., 2021; Feng et al., 2023).

## D   RELATED WORK

### D.1   DENOISING FOR MOLECULAR PRE-TRAINING

Denoising as a self-supervised learning task originates from denoising generative models in computer vision (Vincent et al., 2008). In molecular pre-training, it refers to corrupting and reconstructing the 3D structure of the molecule. Denoising is a self-supervised learning task designed

Table 14: Hyperparameters for fine-tuning on QM9.

| Parameter | Value or description |
|---|---|
| Train/Val/Test Splitting | 110000/10000/remaining data |
| Batch size | 128 |
| Optimizer | AdamW |
| Warm up steps | 10000 |
| Max Learning rate | 0.0004 |
| Learning rate decay policy | Cosine |
| Learning rate factor | 0.8 |
| Cosine cycle length* | 300000 (500000) |
| Network structure | Geometric Equivariant Transformer |
| Head number | 8 |
| Layer number | 8 |
| RBF number | 64 |
| Activation function | SiLU |
| Embedding dimension | 256 |
| Head | |
| Standardize | Applied according to (Thölke & Fabritiis, 2022) |
| AtomRef | |
| Label weight | 1 |
| Noisy Nodes denoise weight | 0.1(0.2) |
| Coordinate noise scale | 0.005 |

Table 15: Random seeds have little effect on the finetuning of QM9 and MD17.

| | homo(QM9) | lumo(QM9) | Aspirin(MD17) |
|---|---|---|---|
| seed=0 | 14.0 | 12.3 | 0.1714 |
| seed=1 | 13.6 | 12.3 | 0.1740 |
| seed=2 | 13.3 | 12.2 | 0.1744 |
| Mean(Standard deviation) | 13.63(0.35) | 12.27(0.06) | 0.1733(0.0016) |

specifically for 3D geometry data and achieves outstanding results in many downstream tasks for 3D molecules (Zhou et al., 2023; Feng et al., 2023).

The existing denoising methods mainly differ in the noise distribution and denoise tasks. Uni-Mol (Zhou et al., 2023) adds uniform noises of $[-1\text{Å}, 1\text{Å}]$ to the random $15\%$ atom coordinates. The model is trained to recover the correct atom coordinates and pair distance. They combine denoising with atom-type masking to make the masking task more challenging.

Coordinate denoising (Coord) (Zaidi et al., 2023) adds Gaussian noise to atomic coordinates of equilibrium structures and trains the model to predict the noise from the noisy input. They establish the equivalence between coordinate denoising and force field learning. Transformer-M (Luo et al., 2023) utilizes Coord to train the 3D model they proposed.

To capture the anisotropic molecular probability, fractional denoising (Frad) (Feng et al., 2023) proposes to add a hybrid noise on the dihedral angles of rotatable bonds and atomic coordinates, and fractionally denoise the coordinate noise. In this specially designed denoising task, the physical interpretation of learning force field also holds.

Compared to the aforementioned methods, our work most closely aligns with physical principles because our energy function better describes the true molecular energy landscape. This leads to a more realistic molecular force field and sampling distribution that is beneficial for representation learning.

On the other hand, to make the molecular energy invariant to rotation and translation, 3D-EMGP (Liu et al., 2023b) denoises the Gaussian noise on the pairwise atomic distances and SE(3)-DDM (Jiao et al., 2023) exploits the Riemann-Gaussian distribution for coordinate denoising. Our

method naturally satisfies the symmetric prior because our energy function is defined on bond length, bond angle and dihedral angle, which are invariant to rotation and translation.

## D.2 3D Molecular Modeling in Relative Coordinates

The geometric information contained in 3D conformers is crucial for molecular representation learning. Though most 3D structures are represented in Cartesian coordinates, recently many works have focused on utilizing 3D information in relative coordinates i.e. bond length, bond angle, torsion angle, also called as internal coordinates or local coordinates. The relative coordinates capture the complete geometry of atomic structures and are widely used because they are invariant to rotation and translation, making them convenient for molecular description in many scenarios (Li et al., 2023).

For one thing, relative coordinates are used to enhance the expressiveness of graph neural networks. For molecular property prediction, SphereNet (Liu et al., 2022) and GemNet (Klicpera et al., 2021) encode bond length, bond angle and dihedral angle information by spherical Bessel functions and spherical harmonics functions. ALIGNN-d (Hsu et al., 2022) encode relative coordinates information by Radial Bessel basis and Gaussian basis and learn representations for optical spectroscopy prediction.

For the other thing, the prediction of relative 3D information is found effective in pre-training task design. ChemRL-GEM(Fang et al., 2021) propose to predict bond lengths and bond angles to describe the local spatial structures. 3D PGT (Wang et al., 2023a) and GearNet (Zhang et al., 2023) also incorporate the prediction of bond length, bond angle and dihedral angle. They differ significantly from BAT-denoising in that their input structures remain unperturbed.

## E Supplementary Background Knowledge

### E.1 Boltzmann Distribution

In statistical physics, the distribution of a collection of identical but distinguishable particles conforms to the Boltzmann distribution Boltzmann (1868). We assume the distribution of the conformations of a molecule satisfies the Boltzmann distribution:

$$p(\boldsymbol{x}) = \frac{1}{Z} e^{-\frac{E(\boldsymbol{x})}{k_B T}}, \tag{41}$$

where $E(\boldsymbol{x})$ is the (potential) energy function, $\boldsymbol{x} \in \mathbb{R}^{3N}$ is the position of the atoms, i.e. conformation, N is the number of atoms in the molecule, $T$ is the temperature, $k_B$ is the Boltzmann constant and $Z$ is the normalization factor. When employing neural networks to fit the energy function or its gradient, the constant $k_B T$ can be absorbed in the energy function, i.e. $p(\boldsymbol{x}) \propto exp(-E(\boldsymbol{x}))$.

### E.2 Classical Intramolecular Potential Energy Functions

Here we provide more information on the classical intramolecular potential according to (Mol, 2020). The classical intramolecular potentials consider $N(N-1)/2$ interactions between pairs of atoms in a molecule with N atoms. According to the pair distance, the interactions fall into four categories: 1-2, 1-3, 1-4 and farther interactions. Specifically, 1-2 interactions denote interactions between covalently bonded atoms. 1-3 interactions denote interactions between atoms i and k, where i-j and j-k are covalently bonded but i-k is not. Similarly, 1-4 interactions denote interaction where the two atoms are separated by a chain of three covalent bonds. Farther interactions refer to 1–5, 1–6, etc. interactions.

$$\begin{aligned} E(\boldsymbol{r}, \boldsymbol{\theta}, \boldsymbol{\phi}) = & \underbrace{\frac{1}{2} \sum_{i \in \mathbb{B}} k_i^B (r_i - r_{i,0})^2}_{\text{1-2 interactions}} + \underbrace{\frac{1}{2} \sum_{i \in \mathbb{A}} k_i^A (\theta_i - \theta_{i,0})^2}_{\text{1-3 interactions}} \\ & + \underbrace{\sum_{i \in \mathbb{T}} k_i^T (1 - cos(\omega_i(\phi_i - \phi_{i,0})))}_{\text{1-4 interactions}} + \underbrace{\sum_{(j,k) \in \mathbb{F}} E_{elec}^{(j,k)} + \sum_{(j,k) \in \mathbb{F}} E_{vdW}^{(j,k)}}_{\text{1-4 and farther interactions}} . \end{aligned} \tag{42}$$

In equation 5, as is rewritten in detail in equation 42 above, the 1-2 interactions are described by bond stretch potentials and the 1-3 interactions are modeled by the angle bend potentials. As for 1-4 interactions, it is modeled by bond torsion potentials, 1–4 electrostatic (elec) and 1–4 van der Waals (vdW) interactions. Farther interactions are described by electrostatic and van der Waals interactions only. $(j, k) \in \mathbb{F}$ denotes atom j and atom k are separated by three covalent bonds or more. The potential energy function in the form of equation 42 together with its modification are widely adopted in molecular simulations. The parameters are obtained through fitting experimental data or quantum chemical calculations.

