# OpenReview forum: "Sliced Denoising: A Physics-Informed Molecular Pre-Training Method"
_ICLR.cc/2024/Conference — ICLR 2024 poster_

### Official Review · Reviewer_nTFT · 2023-10-29

**Soundness:** 3 good
**Presentation:** 3 good
**Contribution:** 3 good
**Rating:** 8
**Confidence:** 5

**Summary:**

The paper proposed a improved denoising pretraining method for 3D GNN in molecular machine learning tasks. The proposed method, called sliced denoising (SliDe), is inspired by classical intramolecular potential theory. SliDe adds different levels of noise to the length of bonds, the magnitude of angles, and the magnitude of torsion angles. Through a series of theoreticall derivation, the authors demonstrate the equivalence of SliDe method and learning the molecular force field. Lastly, the authors show that the SliDe has outperformed previous denoising training methods on the QM9 and MD17 dataset.

**Strengths:**

1. Comparing to previous work such as the Fractional Denoising, the noise designing of this work is a more "physical" formulation because changing either one of the bond length, angle degree, or torsion angle degree does not affect the rest.

2. The writing and proof process are of high clarity.

3. Some efforts to the direction of explainablity are interesting. For example, the correlation coefficient between the learned force field and ground-truth force field (Table 1) is a nice way to quantified the learned force field.

**Weaknesses:**

1. In section 3.1, the claim above Eq. 6 is not correct. Modeling long-range electrostatic (coulomb) interaction is critical in many areas including energy calculation and md simulation. For large system, long-range vdW outside of a cutoff distance may be neglected, but for small molecules in QM9 and MD17, it should not be neglected. When using GNN that takes 3D coordinate and atom type as input approximate energy function, the model should be able to learn those two terms, thus not affecting the approximation of Eq.6. However, the authors should rephrase the sentence. The two citations associated with the sentence does support the claim so they should be removed.

2. The noise design is very import in this work. In the BAT noise (Eq. 9), the parameter vectors are critical but there isn't detailed explaination of them. The authors briefly discussed in the section C.1, but I do think more details and example of those parameter vectors can substentially help reader in understanding the noise design.

3. Missing parenthesis in the second exponential term of Eq. 8. $(\theta_{i} - \theta_{i, 0})^2$

4. I do think the superiority of the SliDe method can be strengthen by more downstream experiments, especially energy prediction. For example, the ANI-1x dataset (www.nature.com/articles/s41597-020-0473-z) is a excellent dataset for such task.

**Questions:**

1. Table 7 is confusing. My understanding is that “Training from Scratch” meaning no pretraining, and Coord and Frad meas pretrained with different method and then fine-tuned on MD17-Aspirin. What does DFT label supervised mean? Isn’t “Training from Scratch” also supervised? The authors should elaborate. The unit of the prediction MAE should also be included in the table.

---

> ### Author Response · Authors · 2023-11-19
>
> We greatly appreciate your expert feedback and your insightful concerns. Your concerns and questions are responded as follows:
>
> >Weakness 1: The claim above Eq. 6 is not correct and the authors should rephrase the sentence. The two citations should be removed.
>
> We appreciate the valuable suggestion. Indeed, electrostatic and van der Waals interactions are important in molecular simulation to maintain the long-range structure of molecules in classical simulations. However, we do not expect the self-supervised pretraining force target to be as accurate as precise computational methods, which would greatly increase the burden of the pretraining task. We strike a balance between complexity and effectiveness, and raise the force field accuray by 42%(as shown in experiment in section 4.1) without substentially increasing the time complexity of pre-training(as shown in responses to ft62 question 1).
>
> As for the explaination for the approximation, we would like to rephase is as follows. "Secondly, we drop the last two terms in order to get a quadratic form of energy function in equation 6. Despite these long-range electrostatic and van der Waals interactions are important in classical simulations, we find the accuracy of the approximated energy function is much higher than existing self-supervised pre-training methods in section 4.1." If you have a better explanation or further suggestions, please feel free to respond us.
>
> >Weakness 2: I think more details and examples of those parameter vectors can substantially help the reader in understanding the noise design.
>
> Thanks for the helpful suggestion. Here we provide some examples of BAT noise in the figure https://anonymous.4open.science/r/SliDeRebuttal-EEC9/STDBat32.jpg. (a)(b) Standard deviations of BAT noise for acetaldehyde. (c) Standard deviations of BAT noise for 2-methylpyridine. We can see that the noise scale can distinguish different bond types and atom types, resulting in better molecular modeling.
> We have added the examples in appendix C.1 in the new version.
>
> >Weakness 3: Missing parenthesis in the second exponential term of Eq. 8.
>
> Thank you for the kind reminder, and we have revised it in the new version.
>
> >Weakness 4: The superiority of the SliDe can be strengthened by more downstream experiments, especially energy prediction, for example, on the ANI-1x dataset.
>
> We add an experiment on ANI-1x energy prediction as below.
> The result is compared with baseline nonequilibrium denoising (Noneq) [1]  that performs coordinate denoising as Coord (Zaidi et al., 2022), while Noneq is pre-trained on ANI-1 and ANI-1x that contain nonequilibrium structures. Noneq also adopts TorchMD-NET as the backbone model.
> SliDe performs better in both pre-train improvement and without pre-train setting, indicating the superiority of SliDe pre-training method over nonequilibrim coordiante denoising as well as the improved backbone model GET over TorchMD-NET.
> | ANI-1x  energy prediction (MAE, kcal/mol) | Noneq | SliDe |
> | --- | --- | --- |
> |  w/o pre-train | 1.50 | 1.362 |
> |  pre-train | 1.01 | 0.786 |
> |  pre-train improvement | 32.7% | 42.3%  |
>
> [1] Denoise Pretraining on Nonequilibrium Molecules for Accurate and Transferable Neural Potentials. J. Chem. Theory Comput. 2023.
>
> >Question 1.1: "What does DFT label supervised mean?"  Isn’t “Training from Scratch” also supervised? The authors should elaborate.
>
> "DFT label supervised" means pre-training by learning the DFT force labels. Besides, your understanding of “Training from Scratch”, "Coord" and "Frad" is exactly right. We add more explanations in appendix B.2 in the revised version.
>
> >Question 1.2:The unit of the prediction MAE should also be included in the table.
>
> Thank you for the useful suggestion, and we have added the unit in the table.

---

> ### Author Response · Authors · 2023-11-19
>
> We have carefully considered your valuable suggestions and have made necessary revisions. The revised content is shown in blue text in the latest version of the paper, which can be found in: https://anonymous.4open.science/r/SliDeRebuttal-EEC9/SLIDE_ICLR2024.pdf
>
> If you have any further questions and feedbacks, please feel free to contact us. If our answer has successfully resolved your issue, we kindly request that you consider raising the rating. Thanks!

---

> > ### Author Response · Authors · 2023-11-21
> > **Anticipating your response**
> >
> > Thank you for your valuable suggestions on how to improve the manuscript. We have provided comprehensive responses to your questions and incorporated more experimental evidence. We are looking forward to your feedback!

---

> > > ### Comment · Reviewer_nTFT · 2023-11-21
> > > **Response to revision**
> > >
> > > Thanks you for answering my questions and concerns point-by-point. I do believe that the revision has elevated the clarity and quality of the manuscript. I'll raise the score.

---

### Official Review · Reviewer_j2Vc · 2023-10-31

**Soundness:** 2 fair
**Presentation:** 3 good
**Contribution:** 3 good
**Rating:** 5
**Confidence:** 4

**Summary:**

This paper proposes a new pre-training method, Slide, that is based on intramolecular potential theory. To lower the computational expense of Slide, the authors introduce a random slicing approach. In addition, a new MLFF architecture is introduced (GET).

**Strengths:**

Strength 1: The writing and motivation of this paper is very clear, with a good description of related works.

Strength 2: Molecular pre-training is an important problem which the authors propose a novel approach for.

**Weaknesses:**

Weakness 1 (Major): The baselines used in this paper are not up to date. In fact, Gemnet (2021) and Nequip (2022) can outperform Slide on basically all of the MD17 and require no pre-training. In addition, the authors claim they set a new state of the art on these benchmarks, which is incorrect.

Weakness 2 (Major): For the downstream task only one random seed is used and the gains over other methods are relatively minor (i.e. compared to Coord and Frad). This makes me doubt that the results are really significant or if they are just due to tuning. I think that multiple random seeds should be reported.

Weakness 3 (Minor): Showing the best result over random seeds in table 1 is kind of strange. I think that the mean result should be shown.

**Questions:**

Question 1: How does your slicing method related to sliced score matching [1]?

[1] Song, Yang, et al. "Sliced score matching: A scalable approach to density and score estimation." Uncertainty in Artificial Intelligence. PMLR, 2020.

Question 2: How does the scale of the pre-training data effect performance?

---

> ### Author Response · Authors · 2023-11-20
>
> We greatly appreciate your expert feedback and your insightful concerns. Your concerns and questions are responded as follows:
>
> >Weakness 1:  Gemnet (2021) and Nequip (2022) can outperform Slide on basically all of the MD17 and require no pre-training. The claim of new state of the arts is incorrect.
>
> First of all, please note that the results of MD17 in Gemnet and Nequip use a different unit (meV/$\mathring{\textnormal{A}}$) than ours (kcal/mol/$\mathring{\textnormal{A}}$). After unifying the unit, SliDe's results in paper surpass Gemnet on 6/8 molecules and NequIP((l= 1) on all 8 molecules in MD17, as shown below.
>
> Furthermore, we find the hyperparameter WoFE indicating weight of force over energy in loss functions is important for MD17 dataset[1]. For fair comparisons with Gemnet, we add a setting that uses the same WoFE as Gemnet that notably raises performance. (Due to the time limit, we have completed the test for five molecules so far, and it is enough to show SliDe's competence.)
>
> We also notice that NequIP reports two versions of results on MD17: l=1 and l=3, where l is the order of spherical harmonics. Recent works[1][2] choose to only compare with NequIP(l=1) because NequIP with high order spherical harmonics is substantially slower[2]. For the same reason,  we compare with NequIP(l=1).
>
> Finally, our main contribution is the pre-training method which is applicable to many backbone models. Therefore, SliDe can actually apply to Gemnet, Nequip and other competitive backbone models.
> | Force prediction MAE (kcal/mol/ $\mathring{\textnormal{A}}$) | WoFE | ASP | BEN | ETH | MAL | NAP | SAL | TOL | URA |
> | --- | --- | --- | --- | --- | --- | --- | --- | --- | --- |
> | Gemnet-T | 1000 | 0.219  | **0.145** | **0.085**  | 0.155  | 0.055  | 0.127 | 0.060 | 0.097  |
> | GemNet-Q | 1000 | $\underline{0.217}$  | **0.145** | $\underline{0.088}$  | 0.159  | 0.051  | 0.125  | 0.060 | 0.104  |
> | NequIP((l= 1) | 1000  | 0.348 | 0.187 | 0.208 | 0.337 | 0.097 | 0.238 | 0.101 | 0.173 |
> | SliDe(in paper) | 4 | **0.174**   | $\underline{0.169} $ | $\underline{0.088} $ | $\underline{0.154 }$ | $\underline{0.048}$ | $\underline{0.101}$ | $\underline{0.054}$ | $\underline{ 0.083 }$  |
> | SliDe | 1000 | - | - | **0.085** | - | **0.043** | **0.095** | **0.049** | **0.079** |
>
> [1] Spherical Message Passing for 3D molecular Graphs, ICLR2022
>
> [2] TorchMD-NET: Equivariant Transformers for Neural Network based Molecular Potentials, ICLR2022
>
> >Weakness 2: I think that multiple random seeds should be reported.
>
> All the results reported in paper use the same random seed 1. This follows the practice of previous literature[1][2][3], because the performance is quite stable for random seeds on QM9 dataset, as shown in [3][4][5]. However, to further validate the stability and to answer this question, we conduct experiments with three different seeds (seed=0,1,2) on homo and lumo tasks in QM9 and Aspirin force prediction in MD17. The results are given in the table below, showing that the performance is quite stable across these tasks.
> | MAE | homo(meV) | lumo(meV) | ASP(kcal/mol/ $\mathring{\textnormal{A}}$) |
> | --- | --- | --- | --- |
> | seed=0 | 14.0 | 12.3 | 0.1714 |
> | seed=1 | 13.6 | 12.3 | 0.1740 |
> | seed=2 | 13.3 | 12.2 | 0.1744 |
> | Mean(Standard deviation) | 13.63(0.35)  | 12.27(0.06) | 0.1733(0.0016) |
>
>
>
> [1]Shikun Feng, Yuyan Ni, Yanyan Lan, Zhi-Ming Ma, and Wei-Ying Ma. Fractional denoising for
> 3D molecular pre-training.
>
> [2]Shengchao Liu, Hongyu Guo, and Jian Tang. Molecular geometry pretraining with SE(3)-invariant
> denoising distance matching.
>
> [3]Yi Liu, Limei Wang, Meng Liu, Yu-Ching Lin, Xuan Zhang, Bora Oztekin, and Shuiwang Ji. Spheri
> cal message passing for 3D molecular graphs.
>
> [4]Kristof Sch ̈utt, Oliver Unke, and Michael Gastegger. Equivariant message passing for the prediction
> of tensorial properties and molecular spectra.
>
> [5]Kristof T Sch ̈utt, Huziel E Sauceda, P-J Kindermans, Alexandre Tkatchenko, and K-R M ̈uller.
> Schnet – a deep learning architecture for molecules and materials.
>
> >Weakness 3: "Showing the best result over random seeds in table 1 is kind of strange."
>
> As discussed in the response to weakness 2, we do not select the best result over random seeds. All the results in our submission are from seed=1.

---

> ### Author Response · Authors · 2023-11-20
>
> >Question 1: How does your slicing method related to sliced score matching?
>
> The similarity is that they are both random slicing method which uses random projection to reduce computational cost. The word "slicing" both originates from Sliced Wasserstein distance. The difference is that the slicing technique is applied to different targets. In sliced score matching, the technique is used to reduce the computation of the gradient of the score matrix, while it is used to reduce the computation of the Jacobi matrix of the coordinate transformation function in our paper.
>
> >Question 2: How does the scale of the pre-training data affect performance?
>
> We randomly extract 10W, 50W, 100W data from PCQM4Mv2 dataset. We pre-train on these subsets and finetune on homo(QM9). The results are provided below. We find more pre-train data contributes to better downstream performance. Meanwhile, SliDe is able to achieve competitive results when data is relatively scarce, such as 50W.
> | Number of pre-training data | homo(MAE,meV) |
> | --- | --- |
> | w/o pretrain | 17.6 |
> | 10W | 16.05 |
> | 50W | 14.53 |
> | 100W | 14.21 |
> | ~300W(whole) | 13.6 |
>
>
>  We have carefully considered your valuable suggestions and have made necessary revisions. The revised content is shown in blue text in the latest version of the paper, which can be found in: https://anonymous.4open.science/r/SliDeRebuttal-EEC9/SLIDE_ICLR2024.pdf
>
> If you have any further questions and feedbacks, please feel free to contact us. If our answer has successfully resolved your issue, we kindly request that you consider raising the rating. Thanks!

---

> > ### Author Response · Authors · 2023-11-21
> > **Anticipating your response**
> >
> > Thank you for your valuable suggestions on how to improve the manuscript. We have also included additional downstream results on ANI-1x to further emphasize the superiority of SliDe. We kindly request that you let us know if there are any remaining concerns that we can address to ensure your satisfaction with the revisions and potentially raise the score. We are looking forward to your feedback!

---

> > > ### Comment · Reviewer_j2Vc · 2023-11-22
> > > **Reviewer Response**
> > >
> > > Thank you for the comprehensive response. I still have a few questions.
> > >
> > > 1. With regards to the weakness 1, the authors' claims that they beat the state of the art should be removed for all molecules where they do no truly beat the state of the art, including Nequip with higher order harmonics. From what I can the tell the authors achieve SOTA on two MD17 molecules: uracil and toluene. I do not think the argument that Nequip should be excluded because it is slower than Slide is really fair, since Slide does pre-training. I do not think that not achieving state of the art is detrimental to this paper, but the authors should correctly report results.
> > >
> > > 2. Thank you for running more seeds. My concern was more regarding whether the improvements over coord and frad on MD17 are statistically significant? It seems like a lot of the gains are very small, so it is hard to tell if these results happen because slide has more tuning. I am not sure if displaying the average MAE across molecules would make the gain of slide clearer. If Slide does achieve significant gains over Coord and Frad, I think that the authors should include more discussion on why it does not.

---

> > > > ### Author Response · Authors · 2023-11-22
> > > >
> > > > Thank you for your suggestions, which make the presentation of the results in our paper more rigorous. The latest version of the paper is updated in the link https://anonymous.4open.science/r/SliDeRebuttal-EEC9/SLIDE_ICLR2024.pdf. The detailed responses are listed below:
> > > >
> > > > Towards Q1: We have revised the imprecise new SOTA statement in section 4.2.2 to "we outperform or achieve comparable results as compared with recent baselines."  Upon double-checking the paper, we confirm that there are no more SOTA statements about MD17.
> > > >
> > > > Towards Q2: To assess the statistical significance of SliDe over Frad and Coord, we conducted two-sample one-sided T-tests on the results of the three models on MD17, as shown in the table below. Since T-tests assume that the data come from normal distributions, we normalized the raw mean absolute error (MAE) before conducting the T-tests. The p-values of the tests are smaller than 0.05, indicating that SliDe achieves a statistically significantly smaller MAE than Frad and Coord.
> > > >
> > > > | Null hypothesis | Alternative hypothesis | Test decision | P value |
> > > > | --- | --- | --- | --- |
> > > > | mean(SliDe)=mean(Frad) | mean(SliDe)<mean(Frad) | Reject null hypothesis | 0.0171 |
> > > > | mean(SliDe)=mean(Coord) | mean(SliDe)<mean(Frad) | Reject null hypothesis | 0.0146 |
> > > >
> > > > If you have any further questions or feedback, please don't hesitate to contact us. We appreciate your time and effort in raising your concerns. If our response has addressed your issue, we would be grateful if you could consider upgrading our rating. Thank you！

---

> > > > ### Author Response · Authors · 2023-11-23
> > > > **Anticipating Your Reply as Discussion Deadline Approaching**
> > > >
> > > > We would like to express our sincere gratitude for dedicating time to review our paper. We have carefully responded to your questions and made revisions accordingly.  As the discussion deadline approaches, we would like to inquire if our responses have adequately addressed your questions and concerns. We are more than willing to address any concerns to ensure a comprehensive resolution and potentially promote your evaluation. Thank you for your time and expertise.

---

### Official Review · Reviewer_mZU2 · 2023-11-01

**Soundness:** 3 good
**Presentation:** 2 fair
**Contribution:** 2 fair
**Rating:** 5
**Confidence:** 2

**Summary:**

The paper proposes a novel approach to molecular pre-training called Sliced Denoising (SliDe) that leverages physical principles to improve molecular property prediction. The authors introduce a new noise distribution strategy that improves sampling over conformations and a denoising task that learns the force field of the energy function. They evaluate SliDe on benchmark datasets QM9 and MD17 and show that it outperforms traditional baselines in terms of physical consistency and molecular property prediction accuracy.

**Strengths:**

The main contribution of the paper, the pre-training algorithm, seems to be extremely relevant to the drug discovery domain with the magnitude of improvement achieved across all benchmark tasks. The paper could be of great interest to scientists in this area.

The pre-training method introduced leverages physical principles and is more interpretable. In addition, the experimental results seem very thorough.

**Weaknesses:**

While the paper is interesting, and makes an important contribution to the field of drug discovery, I would like to raise the question of if ICLR is the correct venue for this submission. This is an important area, and there will be a subset of audience interested in the field, but I would assume that a broader audience will have trouble understanding the paper due to the about of domain knowledge involved. I will leave it to the AC to determine this.

I found the paper hard to read and understand due to the amount of domain knowledge involved. I understand that it is not possible to introduce all the background information in 8 pages, but I would urge the authors to rewrite the paper in a more accessible way for non-domain but ML experts.

**Questions:**

NA

---

> ### Author Response · Authors · 2023-11-20
>
> >I would like to raise the question of if ICLR is the correct venue for this submission. I would assume that a broader audience will have trouble understanding the paper due to the amount of domain knowledge involved.
>
> Response: We believe our paper is very suitable for ICLR for the following reasons.
> - First of all, AI for drug discovery has become an important area in machine learning. To our knowledge, there are around 41 papers about molecules and proteins accepted by ICLR2023, accounting for 2.9% of the total accepted papers. It is in the same scale as federated learning, an important branch of the machine learning field, having 44 papers accepted. Besides, there are around 130 papers about molecules or proteins submitted to ICLR2024.
> - Additionally, many of our pre-training and property prediction baselines are ICLR conference papers: Coord(ICLR2023), SE(3)-DDM(ICLR2023), Uni-mol(ICLR2023), Transformer-M(ICLR2023), SphereNet(ICLR2022), ET(ICLR2022), DimeNet(ICLR2020).
> - Finally, our paper is closely relevant to the topics of "self-supervised representation learning" and "applications to physical sciences (physics, chemistry, biology, etc.)"  among the Subject Areas of ICLR2024.
>
> For the convenience of a broader audience, we provide a supplementary introduction to the involved background knowledge in Appendix E in the new version, which can be found at: https://anonymous.4open.science/r/SliDeRebuttal-EEC9/SLIDE_ICLR2024.pdf
>
> If you have any further questions and feedbacks, please feel free to contact us. If our answer has successfully resolved your issue, we kindly request that you consider raising the rating. Thanks!

---

### Official Review · Reviewer_ft62 · 2023-11-10

**Soundness:** 3 good
**Presentation:** 4 excellent
**Contribution:** 3 good
**Rating:** 8
**Confidence:** 3

**Summary:**

the paper introduces "sliced denoising" (slide), a novel molecular pre-training method that enhances the physical interpretation of molecular representation learning. traditional denoising methods, though physically interpretable, can suffer from inaccuracies due to ad-hoc noise design, leading to imprecise force fields. slide addresses this by utilizing classical mechanical intramolecular potential theory, leading to a 42% improvement in force field estimation accuracy over existing methods.

**Strengths:**

1. innovative approach: slide introduces an innovative noise strategy (bat noise) and a random slicing technique. this approach significantly enhances the accuracy of force field estimations, making it a pioneering method in the field.

1. alignment with physical principles: the method closely aligns with classical mechanical intramolecular potential theory. it appears to improve the realism of molecular representations as well as help that learned representations are physically interpretable, a critical aspect in molecular sciences.

1. empirical validation: slide demonstrates empirically strong results in force field estimation accuracy and downstream task performance on benchmark datasets qm9 and md17.

1. methodology: the paper combines theoretical soundness with methodological innovations effectively. the use of a quadratic energy function approximation and the consequent noise strategy is interesting.

1. network architecture integration: integrating a transformer-based network architecture that encodes relative coordinate information is a notable strength. this architectural choice complements the novel denoising method, enhancing its adaptability to other works using transformer backbones.

**Weaknesses:**

1. computational complexity: while the random slicing technique addresses computational challenges associated with jacobian matrix estimation, the overall computational demand and efficiency, especially in large-scale applications, are not comprehensively addressed​​​​​​.

1. robustness to noisy data: the robustness of slide to noisy or imperfect real-world data is not thoroughly examined. this aspect is crucial for practical applications where data quality can vary significantly​​.

**Questions:**

1. regarding computational efficiency: can the authors provide more details on the computational requirements of slide, especially when applied to large molecular datasets? how does its computational efficiency compare to existing methods?

2. on generalizability and applicability: what are low-hanging fruits to test the generalizability of slide to other types of geometric data or applications beyond molecular science? how might the method need to be adapted for such scenarios?

3. empirical validation across diverse datasets: could the authors elaborate on potential plans to validate slide on a broader range of datasets, particularly those that may present different challenges than qm9 and md17, such as des15k or oc20 as in the coord paper https://arxiv.org/abs/2206.00133?

a curiosity question:
1. dependence on equilibrium structures: the method's reliance on equilibrium structures, to be clear same as most other methods in this space, for training may limit its effectiveness in scenarios where such structures are not readily available or accurate. are there ways to advance molecular representation learning in such a setting?

---

> ### Author Response · Authors · 2023-11-19
>
> We greatly appreciate your expert feedback and your insightful concerns. Your concerns and questions are responded as follows:
>
> >Weakness 1 & question 1: (1)Can the authors provide more details on the computational requirements of slide, especially when applied to large molecular datasets?
>
> Please note that all the terms in the loss function (eq.17)  except for $GNN_\theta(x)$, are calculated in data preprocessing. As the deep learning package (specifically PyTorch in this case) generates multiple workers in the DataLoader through parallel processes, each responsible for loading and processing individual data samples before adding the processed batches to a queue, this approach ensures that the time-consuming training process remains unimpeded, as long as the queue is consistently filled with batches. Therefore, it will not become the bottleneck of the training process, as $GNN_\theta(x)$ is what takes most of the time.
>
> The computational complexity of the regression target for one molecule is $O(N_v\times\max$ { N,M,F }) for SliDe, where N is the number of atoms, M is the dimension of relative coordinates (total number of bond lengths, bond angles, and bond torsions), F is the complexity of coordinate transformation between Cartesian coordinates and relative coordinates.
>
> Two factors affecting the complexity is $N_v$ and $F$. As for coordinate transformation, the complexity $F=O(\max$ { $N,M$ })[1], which is executed efficiently by RDKit. As for the $N_v$ times sampling procedure, in fact, for every $i\in\{1,\cdots,N_v\}$, the regression target can be calculated in parallel. Therefore the time complexity above can be further reduced to $O(\max${N,M,F}).  This makes pre-training on large molecular datasets possible.
>
> [1]On Updating Torsion Angles of Molecular Conformations, Journal of Chemical Information and Modeling 2005
>
> >(2)How does its computational efficiency compare to existing methods?
>
> As for comparison to existing method, SliDe and Frad are theoretically in the same scale. The computational complexity of the regression target for one molecule is $O(\max${ $M_{rb},N,F_{rb}$ }) for Frad, where $M_{rb}$ is the number of rotatable bonds and $F_{rb}$ is the coordinate transformation between Cartesian coordiantes and diheral angles of rotatable bonds. It is in the same scale to SliDe's $O(\max${N,M,F}).
>
> In practise, although we do not process all $N_v$ targets in parallel and incorporate edge update in network architecture, the training time of Frad and SliDe does not vary significantly. Frad pre-training takes 1d1h14m on 8 NVIDIA A100 GPU, and SliDe pre-training takes 1d17h1m on 8 Tesla V100 GPU.
>
> >Question2: Generalizability and applicability to other types of data.
>
> Except for small molecules, our SliDe can potentially be used in pre-training for protein and material. To be specific, SliDe can replace coordinate denoising in existing protein pre-training method[1][2] and replace force label prediction in material pre-training method[3]. The parameters in the energy function should be adapted for these scenarios and they can be obtained from AMBER Force Field[4].
> As a consequence, SliDe can expand its applications in a series of downstream tasks such as binding affinity prediction, structure-based virtual screening, and material property prediction.
>
> [1] Uni-mol: A universal 3D molecular representation learning framework, ICLR2023
>
> [2] General-purpose Pre-trained Model Towards Cross-domain Molecule Learning, submitted to ICLR2024
>
> [3]From Molecules to Materials: Pre-training Large Generalizable Models for Atomic Property Prediction, submitted to ICLR2024
>
> [4]https://ambermd.org/AmberModels.php

---

> ### Author Response · Authors · 2023-11-19
>
> >Question 3: Elaborate on potential plans to validate slide on a broader range of datasets that present different challenges than qm9 and md17.
>
> We plan to evaluate SliDe on larger number and size of molecules. To this end, we choose to test SliDe on MD22[2] and ANI-1x[3] datasets, whose comparison to MD17 and QM9 is shown below. ANI-1x provides large numbers of molecules with multiple equilibrium and nonequilibrium conformations. MD22 contains larger and diversified molecules than QM9 and MD17, including protein, lipids, carbohydrates, nucleic acids and supramolecules.
>
> | Dataset | Molecule size | Number of molecules  | Number of conformations |
> | --- | --- | --- | --- |
> | QM9    | ∼18 (3-29)  | 133,885 | 133,885 |
> | MD17    | ∼13 (9-21)  | 8 | 3611115  |
> | MD22 | ∼67 (42-370) | 7 | 223422 |
> | ANI-1x | ∼ 15 (4-63) | 63865 | 5496771 |
>
> - Baseline: The result is compared with baseline nonequilibrium denoising (Noneq) [1]  that performs coordinate denoising as Coord (Zaidi et al., 2022) in our paper, while Noneq is pre-trained on ANI-1 and ANI-1x that contain nonequilibrium structures. Noneq also adopts TorchMD-NET as the backbone model.
> - ANI-1x:  The result of experiment on ANI-1x energy prediction is shown below. SliDe performs better in both pre-train improvement and without pre-train setting, indicating the superiority of SliDe pre-training method over nonequilibrim coordiante denoising as well as the improved backbone model GET over TorchMD-NET.
> | ANI-1x  energy prediction (MAE, kcal/mol) | Noneq | SliDe |
> | --- | --- | --- |
> |  w/o pre-train | 1.50 | 1.362 |
> |  pre-train | 1.01 | 0.786 |
> |  pre-train improvement | 32.7% | 42.3%  |
>
> - MD22: Due to the time limit, we only test on nucleic acids, consisting of ATAT(60 atoms) and ATATCGCG(118 atoms). The data split follows [1] (train:validation:test=8:1:1). Since Noneq focus on energy prediction and does not report force prediction result, we reevaluate it with both energy and force tasks as a fair comparison. The results below demostrate SliDe is consistently effective for molecules with varied type and size, indicating the ability to apply to broader range of molecules.
> | MD22 MAE (kcal/mol/$\mathring{\textnormal{A}}$)  | ATAT(60 atoms)  |  | ATATCGCG(118 atoms)  |  |
> | --- | --- | --- | --- | --- |
> |  | Force | Energy | Force | Energy |
> | Noneq w/o pre-train | 0.2744 | 0.1098 | 0.7581 | 0.2993 |
> | Noneq  w/ pre-train | 0.2194 | 0.0776 | 0.6574 | 0.2764 |
> | SliDe w/o pre-train | 0.0700 | 0.1021 | 0.0873 | 0.1049 |
> | SliDe w/ pre-train | 0.0444 | 0.0872 | 0.0596 | 0.0904 |
>
> [1] Denoise Pretraining on Nonequilibrium Molecules for Accurate and Transferable Neural Potentials. J. Chem. Theory Comput. 2023.
>
> [2]  Accurate global machine learning force fields for molecules with hundreds of atoms, Science Advances, 2023.
>
> [3] The ANI-1ccx and ANI-1x datasets, coupled-cluster and density functional theory properties for
> molecules. Scientific data, 2020.
>
>
> >Weakness 2 & Curiosity question: (1) Dependency on equilibrium structures
>
> We find pre-training with inaccurate conformation also works. Due to the time limit, we extract subsets of 10W and 50W molecules from PCQM4Mv2 dataset and generate inaccurate conformations by RDKit (the conformation is optimized by MMFFOptimizeMoleculeConfs function provided by RDKit, but still not as accurate as DFT). We pre-train on the dataset we construct and test them on homo(QM9), as shown below.
> | homo(MAE,meV) | PCQ(DFT) | PCQ(RDKit) |
> | --- | --- | --- |
> | 10W | 16.05 | 16.13 |
> | 50W | 14.53 | 15.39 |
>
> Compared with training from scratch MAE=17.6 meV, pre-training with RDKit data is notably effective. Compared with pre-training with accurate conformations, less accurate conformations compromise the performance. Overall, SliDe is robust to inaccurate conformations, revealing the potential of SliDe in larger scale of pre-training.
>
> >(2) Are there ways to advance molecular representation learning in such a setting?
>
> Although this issue is beyond the scope of this article, it is an academic issue that we believe is worth exploring. We deem the force and energy labels of the nonequilibrium conformation may be helpful to model the local energy function landscape and design a reasonable pre-training task[4][5]. But the specific usage of labels and task design still needs more research.
>
> [4] May the Force be with You: Unified Force-Centric Pre-Training for 3D Molecular Conformations, Neurips 2023
>
> [5] Generalizing Denoising to Non-Equilibrium Structures Improves Equivariant Force Fields, submitted to ICLR2024

---

> ### Author Response · Authors · 2023-11-19
>
> We have carefully considered your valuable suggestions and have made necessary revisions. The revised content is shown in blue text in the latest version of the paper, which can be found in: https://anonymous.4open.science/r/SliDeRebuttal-EEC9/SLIDE_ICLR2024.pdf
>
> If you have any further questions and feedbacks, please feel free to contact us. Thanks!

---

> > ### Comment · Reviewer_ft62 · 2023-11-21
> > **rebuttal reply**
> >
> > dear authors,
> >
> > thank you very much for your extensive answers!
> >
> > i maintain my rating of 8 (accept) and want to congratulate you on an interesting paper.
> >
> > in good spirits,
> >
> > reviewer  ft62

---

### Meta-Review · Area_Chair_iTyv · 2023-12-05

**Metareview:**

The paper contributes to the quickly growing field of representation learning for molecules. The main challenge in this field is finding novel and enriching learning signals. In contrast to NLP or vision, there are no natural self-supervised tasks that achieve dramatic performance boosts. The paper makes a convincing argument that the proposed method improves over previous versions of denoising-based pretraining tasks, which makes one optimistic that it is complementary to other pretraining objectives (such as using chemical reactions or 2D graphs). During the discussion, one of the major concerns related to the accuracy of state-of-art performance claims. The Authors have largely addressed this comment (in my opinion) by adding a comparison with Gemnet. I also agree with their remark that the proposed pretraining objective, which is the main contribution of the paper, can be used in combination with any backbone architecture. All in all, it is my pleasure to recommend acceptance of the paper.

**Justification For Why Not Higher Score:**

Its contribution is not that generic -- it is relatively specific to molecule tasks.

**Justification For Why Not Lower Score:**

See meta-review.

---

### Decision · Program_Chairs · 2024-01-16

Accept (poster)